# Biophysical clocks face a trade-off between internal and external noise resistance

Weerapat Pittayakanchit[1,2†], Zhiyue Lu[1,2†], Justin Chew[3], Michael J Rust[1,2,4], Arvind Murugan[1,2]*

[1]Department of Physics, University of Chicago, Chicago, United States; [2]The James Franck Institute, University of Chicago, Chicago, United States; [3]Medical Scientist Training Program, Pritzker School of Medicine, University of Chicago, Chicago, United States; [4]Department of Molecular Genetics and Cell Biology, University of Chicago, Chicago, United States

**Abstract** Many organisms use free running circadian clocks to anticipate the day night cycle. However, others organisms use simple stimulus-response strategies ('hourglass clocks') and it is not clear when such strategies are sufficient or even preferable to free running clocks. Here, we find that free running clocks, such as those found in the cyanobacterium *Synechococcus elongatus* and humans, can efficiently project out light intensity fluctuations due to weather patterns ('external noise') by exploiting their limit cycle attractor. However, such limit cycles are necessarily vulnerable to 'internal noise'. Hence, at sufficiently high internal noise, point attractor-based 'hourglass' clocks, such as those found in a smaller cyanobacterium with low protein copy number, *Prochlorococcus marinus*, can outperform free running clocks. By interpolating between these two regimes in a diverse range of oscillators drawn from across biology, we demonstrate biochemical clock architectures that are best suited to different relative strengths of external and internal noise.
DOI: https://doi.org/10.7554/eLife.37624.001

*For correspondence:
amurugan@uchicago.edu

†These authors contributed equally to this work

Competing interests: The authors declare that no competing interests exist.

## Introduction

Extracting information from a noisy external signal is fundamental to the survival of organisms in dynamic environments (*Bowsher and Swain, 2014*). From yeast anticipating the length of starvation (*Mitchell et al., 2015*) and bacteria estimating sugar availability (*Tu et al., 2008*), to dictyostelium counting cAMP pulses (*Cai et al., 2014*), organisms must often infer properties of the environment that are masked by noisy irregular fluctuations in order to be well-adapted (*Siggia and Vergassola, 2013*; *Mora and Wingreen, 2010*).

A striking example of regularity in environmental stimuli is the daily day-night cycle of light on earth; organisms from all kingdoms of life use circadian clocks to synchronize - or 'entrain' - in phase to these 24-hour periodic signals in order to anticipate and prepare for future changes in light (*Winfree, 2001*). The most remarkable and well-studied examples of clocks are free running circadian clocks, found in organisms ranging from the cyanobacterium *S. elongatus* to insects, plants and humans. Such clocks use non-linear dynamics to generate self-sustained 24-hr rhythms of a preferred amplitude even in the absence of external driving. Many salient properties have been ascribed to such free running internal rhythms (*Troein et al., 2009*; *Winfree, 2001*).

However, several organisms have only damped clocks or 'hourglass clocks'; their response to daily changes in light is not a self-sustaining oscillation, but rather a physiological program that decays to a steady state over a day. For example, some strains of *P. marinus*, a smaller $0.5\mu m$ cyanobacterium with an estimated $50\times$ smaller protein copy number than *S. elongatus* (*Bryant, 2003*;

**eLife digest** The daily rising and setting of the sun is perhaps the most predictable pattern on Earth. Many organisms, from ancient bacteria to animals and plants, have evolved internal biological clocks to anticipate specific events such as dusk and dawn. However, biological clocks also need to continue working when faced with irregularities – both arising from within the organism and from external factors, such as a passing cloud that darkens the sky.

Some organisms, including humans, have a so-called 'free-running' clock that generates a 24-hour rhythm, and keeps ticking even in the absence of any time triggers. Others, such as certain cyanobacteria, have an 'hourglass' clock that is not self-sustained – rather, these clocks show a simple response to the sunrise (or sunset) that would gradually perish without another sunset (or sunrise).

So far, it has been unclear why organisms have different kinds of clocks and if one type of clock is better suited for some conditions than others. Here, Pittayakanchit, Lu et al. analyzed and compared mathematical models of clocks in a variety of organisms, from cyanobacteria and fungi to plants and animals.

The results revealed that internal and external irregularities put opposing pressures on biological clocks. Free-running clocks are more precise and more robust to external fluctuations, but more susceptible to internal ones. In contrast, hourglass clocks can remain accurate when internal irregularities are high but can be disturbed by external ones.

Biological clocks affect the health of the entire organism and faulty clocks are implicated in numerous diseases. The study of Pittayakanchit, Lu et al. showed that the optimal architecture of a biological clock depends on the balance of irregularities in the external and internal environment of an organism. A next step will be to understand whether an organism can change its clock architecture while the environment changes. A better understanding of how biological clocks are regulated may help us find ways to tune faulty clocks to account for both the external environment and the internal state of an organism.

DOI: https://doi.org/10.7554/eLife.37624.002

*Gutu et al., 2013*; *Holtzendorff et al., 2008*; *Dufresne et al., 2003*; *Kitayama et al., 2003*), appear to have such a damped 'hourglass' clock, despite the clock being constituted from Kai proteins similar to those in *S. elgonatus*.

The potential benefits and drawbacks of these timing systems are not immediately obvious. In particular, it is unclear when an 'hourglass' clock might be sufficient or even preferred over free running clocks.

Here, we compare such classes of clocks when driven by the day-night cycle of light in fluctuating conditions. One source of fluctuations are amplitude fluctuations in the external day-night signal due to weather patterns (*Gu et al., 2001*) or other environmental disturbances. Phase entrainment to such fluctuating environmental signals is a challenge because while amplitude fluctuations are uninformative of phase, an entrainment mechanism looking for dawn-dusk transitions might conflate such amplitude fluctuations with true variations in phase. Biomolecular clocks also face an internal source of fluctuations (*Lestas et al., 2010*), due to various causes like finite copy number effects (*Tsimring, 2014*), bursty transcription, interactions with the cell cycle and cell division (*Teng et al., 2013*). It is clear that the inability to deal with either of these fluctuations will lead to poor phase entrainment, with a host of associated fitness costs in cyanobacteria (*Woelfle et al., 2004*), plants, rodents and humans (*Evans and Davidson, 2013*). However, it is not clear what kinds of clock architecture are best at dealing with internal and external fluctuations and whether these demands are compatible.

We find that free running clocks, based on limit cycle attractors, are a double-edged sword when subject to such internally and externally fluctuating conditions. The flat direction along such continuous limit cycle attractors can selectively project out external amplitude fluctuations while retaining phase information. However, the flat direction along the attractor makes these continuous attractor-based clocks susceptible to internal fluctuations (e.g. low protein copy number [*Potoyan and Wolynes, 2014*]). In contrast, point attractor-based damped clocks are relatively resistant to internal

fluctuations because they have no flat directions. Hence such 'hourglass' clocks out-perform free running clocks at sufficiently high internal noise.

We first demonstrate our results in diverse biochemical oscillators, drawn from the literature (*Leloup et al., 1999*; *Schmal et al., 2014*; *Locke et al., 2005*; *Leloup and Goldbeter, 2003*; *Goldbeter, 1991*; *Goodwin, 1965*; *Gonze and Abou-Jaoudé, 2013*; *Kondepudi and Prigogine, 2014*; *Elowitz and Leibler, 2000*; *Bușe et al., 2009*; *Potvin-Trottier et al., 2016*) on clocks in cyanobacteria, plants and mammals to cell cycle and synthetic oscillators. We complement this detailed network-based study with dynamical systems theory that explains the same trade-off in terms of the broad features common to the diverse models studied here. In all cases, our approach involves systematically deforming the dynamics to interpolate between free running and 'hourglass' clocks and using information theoretic measures to quantify clock performance in the presence of fluctuations.

By continuously interpolating between these clock architectures, our work predicts that a survey of clock systems in different environmental niches will reveal that clock architecture vary systematically with the relative strength of external and internal fluctuations (*Laughlin, 1981*). Further, our work suggests intriguing forward evolution experiments in the lab where the same structured external environment can nevertheless result in distinct regulatory systems, depending on the size of internal fluctuations. Finally, the existence of 'hourglass' clocks are easier to overlook experimentally than free running oscillations. Hence our theoretical demonstration that 'hourglass' clocks have functional benefits in specific conditions highlights the importance of experiments that specifically look for such damped clocks. More broadly, our work highlights the need to experimentally probe regulatory strategies by varying different kinds of noise independently when possible, since the strategies to deal with different kinds of noise are not equivalent and can be in conflict.

## Results

### Free running clocks and damped 'hourglass' clocks

Many organisms like humans and rodents have free running clocks that show self-sustained 24 hr rhythms even in constant dark or light conditions. A particularly simple and well-characterized free running clock is that found in *S. elongatus* where the clock dynamics can be reproduced by the post-translational dynamics of Kai ABC *in vivo* as well. Measuring the phosphorylation state at any one of several sites on KaiC reveals an orderly phosphorylation reaction with a period of 24 hr. As shown in *Figure 1a*, oscillations of a characteristic amplitude are sustained even in constant darkness or constant light, that is, in the absence of a periodic external drive.

Not all organisms have a free-running clock; for example, many insects (*Saunders, 2002*) have damped 'hourglass' clocks that decay to a fixed point under constant light or constant dark conditions but show oscillatory dynamics under day-night cycling (see *Figure 1b*). In fact, a sister cyanobacterial species *P. marinus* has a KaiBC-protein based clock. While the details of this clock are not fully characterized, the clock lacks the KaiA-mediated negative feedback (*Dufresne et al., 2003*; *Holtzendorff et al., 2008*) loop that enables free running oscillations in *S. elongatus*. Consequently, in constant light or dark conditions, the clock's state decays to a distinct day or a night state respectively (*Holtzendorff et al., 2008*).

Thus, both classes of clock show regular oscillations when externally driven. With cloudless day-night cycling, both systems can synchronize in phase with the external signal (i.e., 'entrain') and show distinct clock states at distinct times of the day. In this way, the clock state provides the rest of the cell with an estimate of the time of the day. However, while the free running clock has a natural amplitude relatively independent of the external drive, the damped clock's amplitude is directly set by the external drive.

### External fluctuations

The day-night pattern of light on earth does not resemble the clean square wave shown in *Figure 1a* but is rather subject to large amplitude fluctuations during the day due to weather patterns. Such amplitude fluctuations and their spectrum have been quantified (*Gu et al., 2001*) and also identified as playing a critical role in several studies on the evolution and performance of circadian clocks (*Domijan and Rand, 2011*; *Troein et al., 2009*). The biological impact of such changes

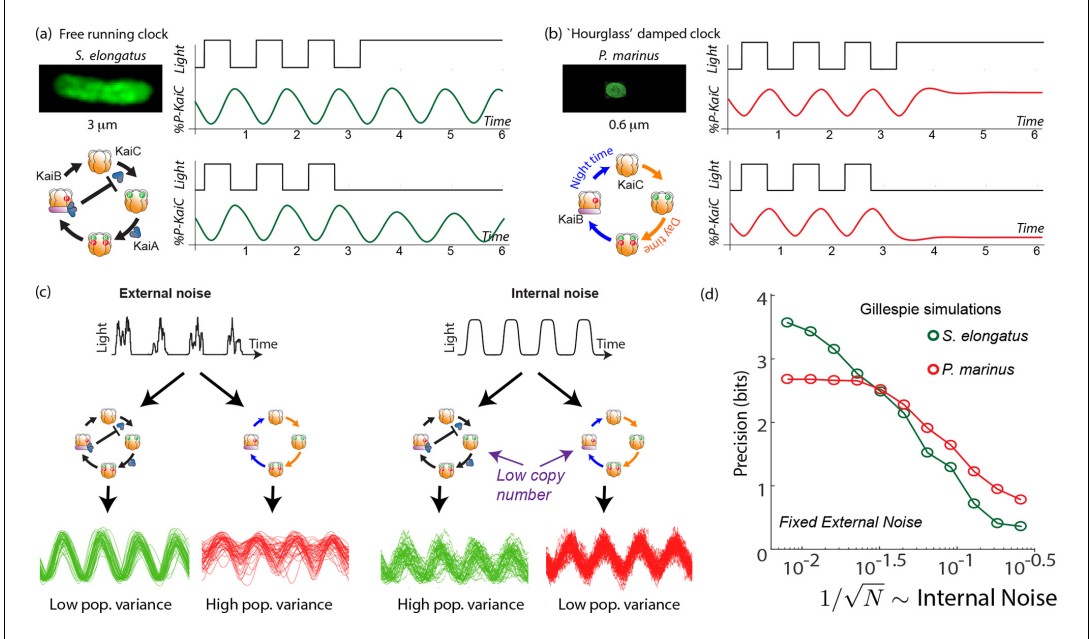

**Figure 1.** Free running clocks and damped 'hourglass' clocks are equally good time-keepers in noiseless conditions but internal and external fluctuations reveal significant differences. (a) Free running circadian clocks, such as the KaiABC protein clock in *S. elongatus*, show rhythms in both oscillating and constant light (top) or dark (bottom) conditions. (b) In contrast, damped circadian clocks, such as that in *P. marinus* which lacks Kai A, show rhythms only in changing light conditions and decay to a fixed state in constant conditions. (c) When subject to external noise (i.e., weather-related amplitude fluctuations in light), simulations of the free running clock show low population variance while the damped clock shows high variance. In contrast, Gillespie simulations with high internal noise due to low copy number of Kai molecule reveals that damped clocks are much more robust than free running clocks. (d) A systematic study of clock precision (i.e., mutual information between clock state and time) at fixed external noise but decreasing Kai protein copy number $N$ reveals that free running clocks are preferred at low internal noise but damped clocks are preferable at sufficiently high internal noise.

DOI: https://doi.org/10.7554/eLife.37624.003

in light intensity in cyanobacteria have been quantified recently (*Teng et al., 2013*). The clock must entrain in phase to the external signal while ignoring such amplitude fluctuations.

## Internal fluctuations

In addition to external fluctuations, circadian clocks also deal with the intrinsically noisy nature of biochemical reactions (*Swain et al., 2002*). Sources of internal noise for clocks include finite copy number effects, bursty transcription, cell division and other sources (*Tsimring, 2014*). In particular, based on their relative sizes (*Dufresne et al., 2003*; *Holtzendorff et al., 2008*; *Bryant, 2003*), *P. marinus* is thought to have far fewer copies of the Kai clock proteins (e.g., $\sim 500$ of KaiC ) than *S. elongatus* ($\sim O(10000)$ copies of KaiC [*Gutu et al., 2013*; *Kitayama et al., 2003*]). Such finite numbers of molecules is known to create significant stochasticity in oscillators in the absence of an external signal (*Potoyan and Wolynes, 2014*).

## Noise resistance of Kai-based clocks

We tested the impact of such external and internal fluctuations on the contrasting clock architectures in *S. elongatus* and *P. marinus* through simulations. We set up explicit Gillespie simulations (*Gillespie, 2007*) of explicit biomolecular models of the post-translational Kai clock that captures the known biochemistry (*Rust et al., 2007*) of *S. elongatus*'s clock and the putative KaiBC clock (*Bryant, 2003*; *Holtzendorff et al., 2008*) in *P. marinus* (*Figure 1*). We do not include transcriptional coupling (*Zwicker et al., 2010*) of the clock here and focus on the core post-translational oscillator. See Appendix 1 for details. The ATP levels in these models (*Pattanayak et al., 2014*) were coupled to an external square wave input of period 24 hr, representing the day-night cycle of light. To model external fluctuations, we modulated the amplitude of the input square wave over a

broad range of frequencies, reflecting the broad frequency spectrum quantified by the Harvard Forest database (*Moore et al., 1996*). To model internal fluctuations, we varied the copy number in these Gillespie simulations.

With only external fluctuations but suppressing internal fluctuations using high copy numbers, we find that the damped oscillator develops a much larger population variance than the free running clock. In contrast, at low copy number (i.e., high internal noise) but with a noiseless external signal, we find the situation is reversed; the free running clock has significantly higher population variance. See *Figure 1c*.

To study this effect quantitatively, we fixed the strength of amplitude fluctuations and increased the internal noise by decreasing the copy number of all Kai proteins in the Gillespie simulation. We measured the resulting mutual information between clock state and objective time of day. (Mutual information is intuitively a measure of population variance along the most informative directions; see Appendix 4 for more.)

We see that the free running clock has higher precision than the damped clock at low internal noise (high copy number). However, as the internal noise is increased, the precision of the free running clock drops earlier and consequently, the damped oscillator has higher precision at sufficiently high internal noise (low copy number). This is shown in *Figure 1d*, where the precision measures the mutual information between the clock state and the time. For a fair comparison, in undriven conditions, different clocks are assumed to lose information at the same rate.

## Noise resistance in other biochemical clocks

While our study here was motivated by the contrasting Kai protein-based clocks in the two cyanobacterial species *S. elongatus* and *P. marinus*, we sought to test the broader validity of our results. Hence we analyzed the internal and external noise resistance in a range of eight well-studied biochemical oscillators in the literature.

These models range from circadian clocks in numerous organisms - *Neurospora* (*Leloup et al., 1999*), *Arabidopsis* (*Schmal et al., 2014*; *Locke et al., 2005*), mammalian cells (*Leloup and Goldbeter, 2003*) - to other oscillators such as cell cycle models (*Goldbeter, 1991*), the Goodwin (*Goodwin, 1965*; *Gonze and Abou-Jaoudé, 2013*) oscillator, the Brusselator (*Kondepudi and Prigogine, 2014*) and the synthetic repressilator (*Elowitz and Leibler, 2000*; *Buşe et al., 2009*) - see *Figure 2*. While the internal noise properties of these oscillators in undriven conditions have been studied before (*Gonze et al., 2002*), here we analyzed the contrasting internal and external noise properties of these oscillators under externally driven conditions. The results are shown in *Figure 2*.

In each case, we set all kinetic parameters to values specified in the original publications and coupled the external driving signal in the way specified in those original publications. As in the Kai clock simulations, the external signal was a square wave with amplitude fluctuations of fixed strength. Finally, we add Langevin noise to the equations to simulate internal noise; when available, we followed the finite volume prescription for rates in the original publications or related papers to set the size of Langevin noise for each reaction. Simulation and model details are in Appendix 2.

These models here all exhibit a Hopf bifurcation as kinetic parameters are tuned. The publications (*Leloup et al., 1999*; *Schmal et al., 2014*; *Locke et al., 2005*; *Leloup and Goldbeter, 2003*; *Goldbeter, 1991*; *Goodwin, 1965*; *Gonze and Abou-Jaoudé, 2013*; *Kondepudi and Prigogine, 2014*; *Buşe et al., 2009*) identified a parameter which when tuned leads to a Hopf bifurcation; that is, on one side of the bifurcation, we find damped oscillations while on the other side, we find free running oscillations of increasing amplitude. We picked three points along this parameter; the green and purple data correspond to free running oscillations of large and smaller natural amplitude relative to the size of the external drive. The red data corresponds to a choice of parameters on the other side of the Hopf bifurcation, that is, to damped oscillations. For the red data, we chose $\mu$ such that the relaxation timescale was comparable to the period of the external driving force, much as in the Kai model of *P. marinus*. The damped oscillator is a useful clock only when the relaxation timescale is comparable to the period.

In each case, we observed the same trade-off as seen in the Kai system; free running oscillations of large amplitude relative to the external drive (green) were most precise when only subject to external noise but are most fragile to internal noise. Damped oscillations in the same oscillator models are more robust and thus are preferable at sufficiently high internal noise. We find that intermediate amplitude free running oscillations show intermediate noise properties. Consequently, we can

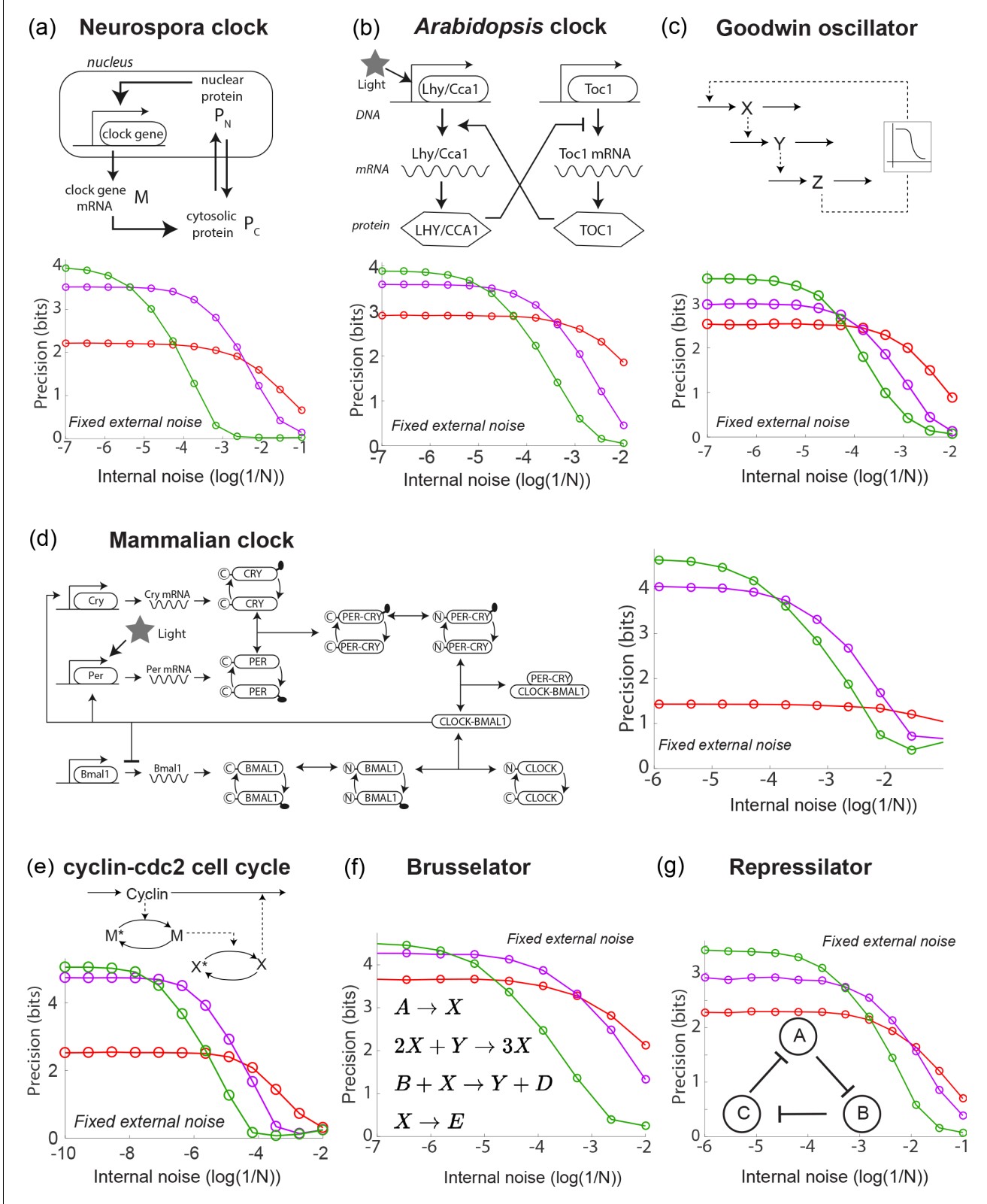

**Figure 2.** A diverse range of biochemical oscillators show the trade-off between resistance to external and internal noise. For each oscillator, the regime (green) of largest free running amplitude relative to the driving strength is most robust to external fluctuations but is most fragile to internal noise. In contrast, damped oscillations (red) are robust to internal noise and thus preferable at sufficiently high internal noise. Regimes (purple) of intermediate free running amplitude are preferred at intermediate internal noise levels. (a–g) Diverse biochemical oscillators from the literature were

*Figure 2 continued on next page*

*Figure 2 continued*

simulated with increasing internal noise $\epsilon_{int} = 1/\sqrt{N}$ while driven by a periodic square wave light signal with fixed strength external noise, using the external coupling and parameters specified in the original publications (*Leloup et al., 1999*; *Schmal et al., 2014*; *Locke et al., 2005*; *Leloup and Goldbeter, 2003*; *Goldbeter, 1991*; *Goodwin, 1965*; *Gonze and Abou-Jaoudé, 2013*; *Kondepudi and Prigogine, 2014*; *Elowitz and Leibler, 2000*; *Buşe et al., 2009*). Clock precision is defined as mutual information between outputs and time. The original publications identified a Hopf bifurcation parameter in these models, with free running oscillations on one side and damped oscillations on the other. Green and purple data correspond to parameter regimes with large and smaller amplitude free running oscillations relative to driving amplitude while the red data corresponds to a damped oscillator. Details in Appendix 2.

DOI: https://doi.org/10.7554/eLife.37624.004

continuously trade-off resistance to internal noise for resistance to external noise by changing the amplitude of free running oscillations relative to the strength of the external drive.

## Dynamical systems theory of noise resistance

We have demonstrated a trade-off between external and internal noise resistance in diverse clocks. While it is possible to trace the origin of this trade-off to specific features of each clock, here, we take a different approach based on dynamical systems theory. Dynamical systems theory has been use to make fruitful general predictions about biological clocks since Winfree's analysis of phase singularities (*Winfree, 2001*). In a similar vein, we use dynamical systems theory to show this trade-off emerges due to basic features of free running and damped clock dynamics and can thus be expected to hold broadly.

## Limit cycle clocks and point attractor clocks

Free running clocks are phenomenologically well-described by a limit cycle attractor, a non-linear oscillator of fixed amplitude (*Winfree, 2001*). While such descriptions have been used for numerous biochemical oscillators, limit cycle dynamics can be experimentally seen in molecular detail for the KaiABC clock in *S. elongatus* as shown in *Figure 3a* (reproduced from [*Leypunskiy et al., 2017*]).

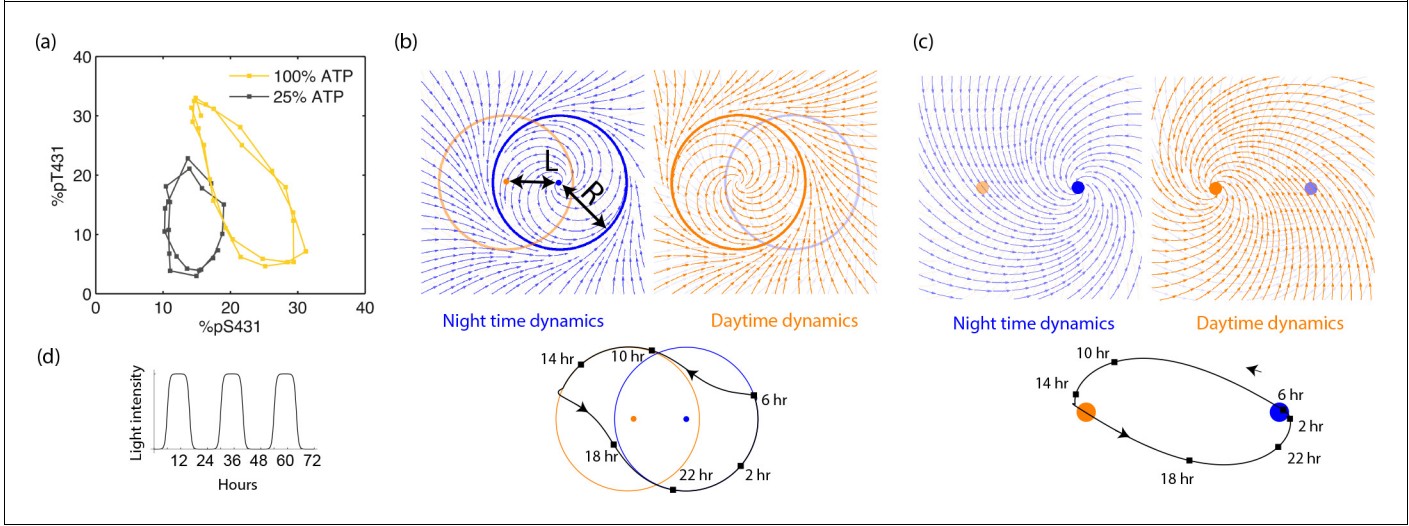

**Figure 3.** Experiments and models of biological clocks show that external driving can be viewed as a switch between distinct day-time and night-time dynamics. (a) Experiments on the Kai system at distinct ATP levels corresponding to day and night conditions reveal limit cycles shifted relative to each other in a phosphorylation space for Kai (reproduced from [*Leypunskiy et al., 2017*]). Similar behavior (*Winfree, 2001*) is seen in models of diverse biochemical oscillators studied in *Figure 2*. (b) We build a minimal model of such driven clocks as a limit cycle of radius $R$ whose center is shifted by a distance $L$ between day and night. In cycling conditions (see signal in (d)), an entrained clock's state executes a trajectory that encompasses both limit cycles as shown (bottom). (c) For damped clocks (*Saunders, 2002*), phenomenology suggests that the day and night limit cycle dynamics are replaced by a point attractor whose position changes between day and night. The relaxation time between the day and night attractors is comparable to ~12 hours, giving rise to the trajectory shown in cycling conditions. (d) The plot shows cycling conditions of light intensities that couple to (b) and (c).
DOI: https://doi.org/10.7554/eLife.37624.005

The clock follows distinct limit cycle dynamics during the day (orange data) and night (black data) (*Leypunskiy et al., 2017*; *Pattanayak et al., 2014*), with the day cycle positioned at higher phosphorylation levels due to a higher ATP/ADP ratio.

The Kai model and indeed the diverse range of biochemical oscillators in *Figure 2* show such a change in the limit cycle between day and night conditions. Here, we build a minimal model of such free-running clocks using circular day and night limit cycles of radius $R$ in a plane. The limit cycle is defined by the dynamics $\tau_{relax}\dot{r} = r - r^3/R^2, \dot{\theta} = \omega$ about its own center; but the center of the limit cycle itself moves along the x axis in *Figure 3b* as $(\rho(t)L, 0)$ where $\rho(t) \in [0, 1]$ is the normalized light intensity level at time $t$ and $L$ is a measure of the physiological changes between day and night (e.g., ATP/ADP ratio change in *S. elongatus*). Thus, for example in *Figure 3b*, the system follows the blue dynamics at night and then after dawn it relaxes to the orange day attractor on a time scale $\tau_{relax}$. Note that $R$ represents the amplitude of free-running oscillations while $L$ represents the strength or amplitude of the external driving signal.

In contrast, damped clocks are phenomenologically well-described by a day-time and a night-time point attractor with slow relaxation dynamics between them (*Figure 3c*), modeled as $\dot{r} = -r/\tau_{relax}, \dot{\theta} = \omega$ about an attractor point whose location varies with current light levels as $(-\rho(t)L, 0)$. Here we assume $2\tau_{relax} \sim 24$ hrs as in *P. marinus* (*Holtzendorff et al., 2008*); if relaxation were faster and completed before the day is over, the clock cannot resolve all times of the day.

Here, we will also consider a family of limit cycle clocks of varying $R/L$ to interpolate between large-$R/L$ limit cycles and point attractors (approximated by $R/L = 0$).

## External noise

We begin with the performance of different clocks in the presence of external intensity fluctuations. Weather patterns cause large fluctuations in the intensity of light over a wide range of time-scales as shown in *Figure 4a*. Much like with biochemical circuits, we subject an in silico population of dynamical system clock models to different realizations of such noisy weather patterns.

When subject to weather fluctuations, we see in *Figure 4b* that the population variance of clock states for limit cycles at given times (purple) is fundamentally limited by the spacing between the day and night limit cycles. Point attractors develop larger overlapping population distributions at different times.

We can geometrically understand the daytime phase variance increase $\sigma^2_{clouds}$ in terms of the phase lag $\Delta\Phi$ due to a single, say $2.4$ hr dark pulse, administered during the day. *Figure 4c* shows that the deviation in trajectory for limit cycle clocks (purple) is fundamentally limited by the presence of a continuous attractor. In contrast, for the point attractor, the trajectory is in free fall towards the night point attractor, with no limit cycle to arrest such a fall. Consequently, the geometrically computed phase shift $\Delta\Phi$ due to the particular dark pulse shown in *Figure 4c* is much smaller for limit cycles ($\Delta\Phi \sim 0.5$ hr for the $R, L$ geometry shown) than for point attractors ($\Delta\Phi \sim 4$ hr) (see Appendix 5).

In fact, this contrast in $\Delta\Phi$ between limit cycles and point attractors holds for dark pulses of any duration and time of occurrence. The contrast is even greater at small $L/R$ since $(\Delta\Phi)^2 \sim (L/R)^2$ for small $L/R$, as shown geometrically in Appendix 5 and confirmed in simulations that average over random weather conditions (*Figure 4d*). Hence, limit cycles are more resistant to external fluctuations than point attractors.

To complete the analysis, we note that phase variance increases additively during the day and falls multiplicatively at dusk (and dawn), that is,

$$\sigma^2 \overset{day}{\to} \sigma^2 + \sigma^2_{clouds} \overset{dusk}{\to} (\sigma^2 + \sigma^2_{clouds})/s^2 \overset{night}{\to} (\sigma^2 + \sigma^2_{clouds})/s^2 \overset{dawn}{\to} (\sigma^2 + \sigma^2_{clouds})/s^4.$$

Solving for steady state phase variance ($\sigma^2 = (\sigma^2 + \sigma^2_{clouds})/s^4$), we obtain

$$\sigma^{2,ext}_{limit\ cycle} \sim \Delta\Phi^2/(s^4 - 1). \tag{1}$$

where we have equated $\sigma^2_{clouds}$ to $\Delta\Phi^2$ for a typical dark pulse Here, $s^2$ represents the variance drop during a dawn/dusk entrainment. As shown in Appendix 5 for external noise (and in *Figure 5* for internal noise), this factor $s$, can be geometrically explained by the slope of the circle map relating

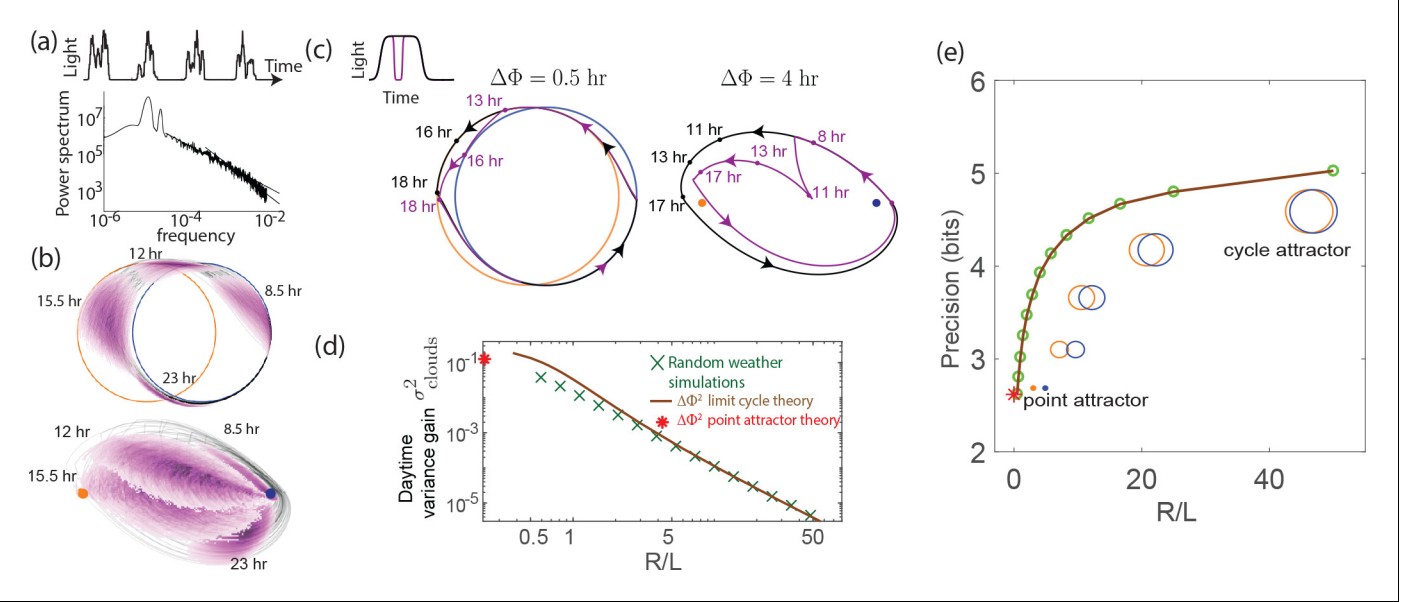

**Figure 4.** External weather-related light fluctuations are filtered out by limit cycle attractors but not by point attractors. (a) Light intensity levels fluctuate on a range of time scales due to weather (power spectrum reproduced from *Gu et al., 2001*). (b) A population of limit cycle clocks of identical fixed geometry, subject to different realizations of weather conditions, show non-overlapping distributions (purple blobs) at different times of the day. Point attractor clocks form larger and more overlapping distributions. (c) A single representative dark pulse of ~2.4 duration causes only a $\Delta\Phi \sim 30$ min phase lag in limit cycles since the trajectory's deviation (purple) is fundamentally bounded by the circular attractor. In contrast, $\Delta\Phi \sim 4$ hr for the point attractor since the trajectory is in free-fall towards the blue night-time attractor. (d) The geometrically computed $\Delta\Phi^2$ phase shift for a dark pulse of any fixed duration and time of occurrence (see Appendix 5) drops rapidly as $(R/L)^{-2}$ for large-$R/L$ limit cycles; this theoretical prediction agrees well with the population variance gain over a day in simulations. (e) Consequently, weakly driven limit cycles (i.e., high $R/L$) can tell time with high precision.

DOI: https://doi.org/10.7554/eLife.37624.006

the two cycles *Leypunskiy et al., 2017*; we find that $s^2 - 1 \sim L/R$ for large-$R/L$ limit cycles. Plugging this and $\Delta\Phi^2 \sim (L/R)^2$ into *Equation1*, we see that $\sigma^2 \rightarrow L/R \rightarrow 0$ for large -$R/L$ cycles.

*Figure 4e* shows that the precision (i.e., mutual information between clock state and time) computed from random weather simulations agrees with this theory; clock precision drops as we interpolate from limit cycles to point attractors by changing $L$ (with equivalent results for changing $R$).

## Internal noise

Internal noise due to finite copy number effects in biochemical networks can be modeled exactly using the Gillespie method used in *Figure 1*. In the context of our dynamical systems model, we follow *Gillespie, 2007* and add Langevin noise to all dynamical variables of the system of strength $\epsilon_{int} \sim 1/\sqrt{N}$, where $N$ is the overall copy number, with the ratios of different species assumed fixed (see Appendix 3). Such a Langevin approach is an approximation *Gillespie, 2007* to the exact Gillespie method used in *Figure 1*.

We simulated a population of clocks in externally noiseless day-night light cycles but with internal Langevin noise. We see in *Figure 5b* that limit cycle populations have significantly higher variance of clock state due to internal noise than point attractors, in contrast to *Figure 4b* with external noise alone.

We can understand the weakness of limit cycle attractor relative to the point attractor in terms of diffusion during day/night balanced by dawn/dusk transitions. The flat direction along the limit cycle attractor cannot contain diffusion caused by the Langevin noise during the day/night; hence the phase variance increases by $\sigma^2 \rightarrow \sigma^2 + \epsilon_{int}^2 T_{day}$ during a day of length $T_{day}$ (and similarly at night).

Dawn and dusk times reduce the phase variance $\sigma^2 \rightarrow \sigma^2/s^2$ as the trajectories originating on, say, the day cycle converge on the night cycle (see *Figure 5c* and *Leypunskiy et al., 2017*; *Monti and Lubensky, 2017*). In fact, we can compute this variance drop $s^2$ entirely through geometric considerations. We define the circle map $\phi = P(\theta)$ as relating originating points $\theta$ near dusk on

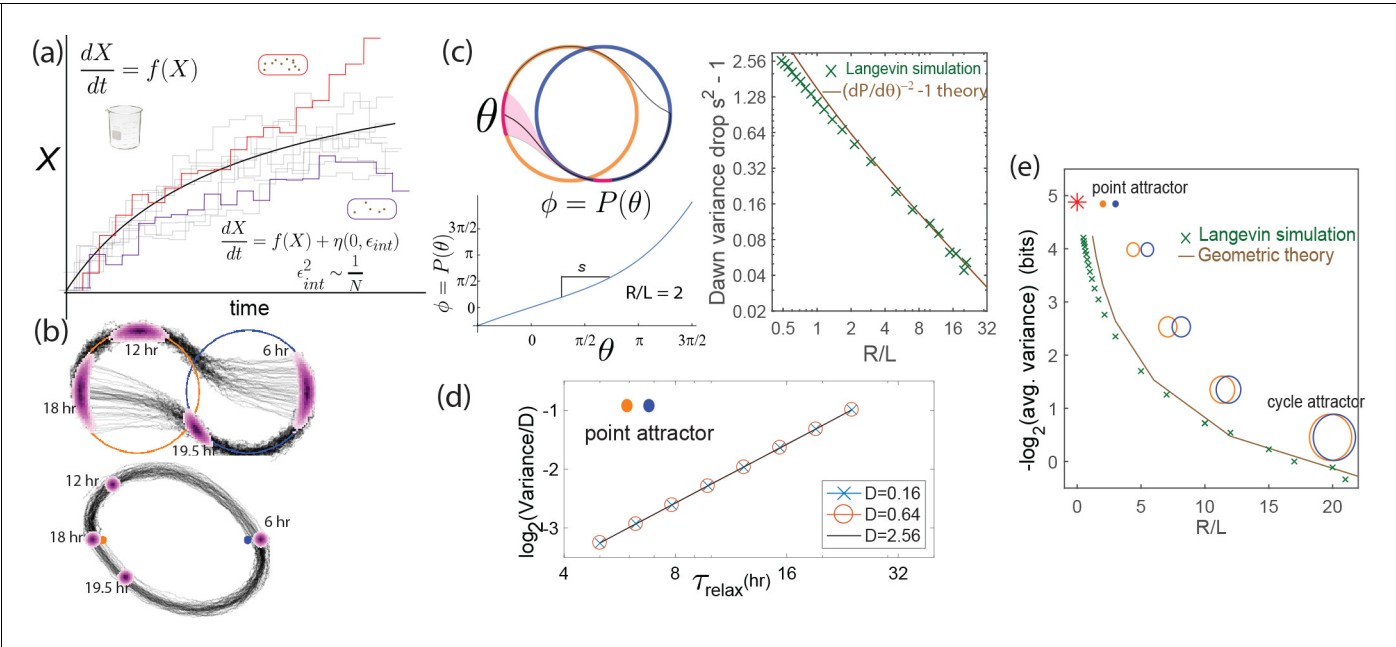

**Figure 5.** Internal fluctuations severely affect continuous attractors but not point attractors. (a) We model fluctuations due to finite copy number $N$ as Langevin noise with mean zero and standard deviation $\epsilon_{int}$, resulting in a diffusion constant $\epsilon_{int}^2 \sim 1/N$ for the clock state. (b) The flat direction of limit cycles cannot contain diffusion, leading to large increases $\epsilon_{int}^2 T_{day}$ in population variance of clock state during each day (and night). In contrast, point attractor dynamics have constant curvature at all times, leading to a constant population variance over time. (c) The variance drops $\sigma^2 \to \sigma^2/s^2$ at dawn and dusk for limit cycles during the off-attractor dynamics between the day and night cycles. As with external noise, the variance drop is predicted by the slope $dP(\theta)/d\theta$ of the circle map between the cycles. This dawn/dusk drop goes to zero for large $R/L$ limit cycles but variance still increases during the day and night. (d) The variance for point attractors is $D\tau_{relax}$, a constant determined by the curvature $\tau_{relax}^{-1}$ of the harmonic potential. (e) Thus, with only internal noise present, the precision of limit cycle clocks increases with increasing separation $L/R$, asymptotically approaching the performance of point attractors.

DOI: https://doi.org/10.7554/eLife.37624.007

the day cycle to final points on the night cycle $\phi$ after relaxation (experimentally characterized in *Leypunskiy et al., 2017*). Then $s^{-1} = dP(\theta)/d\theta$. *Figure 5c* shows that this slope $s^{-1} = dP(\theta)/d\theta$, geometrically computed in the SI, agrees with the dawn/dusk variance drop in Langevin simulations and scales as $s^2 - 1 \sim L/R$ for large $R/L$.

Thus, the population phase variance changes as

$$\sigma^2 \xrightarrow{Day} \sigma^2 + \epsilon_{int}^2 T_{day} \xrightarrow{Dusk} (\sigma^2 + \epsilon_{int}^2 T_{day})/s^2 \xrightarrow{Night} (\sigma^2 + \epsilon_{int}^2 T_{day})/s^2 + \epsilon_{int}^2 T_{day} \xrightarrow{Dawn} ((\sigma^2 + \epsilon_{int}^2 T_{day})/s^2 + \epsilon_{int}^2 T_{day})/s^2.$$

Assuming $T = T_{day} = T_{night}$ and solving for steady-state phase variance $(\sigma^2 = ((\sigma^2 + \epsilon_{int}^2 T_{day})/s^2 + \epsilon_{int}^2 T_{day})/s^2)$, we obtain

$$\sigma_{cycle}^{2,int} \sim \frac{\epsilon_{int}^2 T}{s^2 - 1} \qquad (2)$$

Consequently, as the cycles become large (large $R/L$), the dawn/dusk variance drop vanishes as $s^2 - 1 \sim L/R \to 0$ while diffusion along the flat direction still adds $\epsilon_{int}^2 T$ to the variance during each day and each night; hence large-$R/L$ limit cycles have large $\sigma_{cycle}^{2,int}$ and thus low precision. (Unlike with external noise, internal noise introduces a diffusion length scale and hence changing $L$ and $R$ are not equivalent. To make a fair comparison, we fix $R$ and internal noise while changing $L$ in *Figure 5e*; see Appendix 3 for more detail about other equivalent choices).

Note that *Equation 2* is invalid for strictly undriven clocks (i.e., $s = 1$); such clocks show a variance that increases indefinitely with time. Here, we focus on driven clocks, which always settle to the finite variance given by *Equation 2*.

In contrast, for point attractors, the population variance stays constant during the day-night cycle and is shown to be

$$\sigma_{point}^{2,int} \sim \epsilon_{int}^2 \tau_{relax}$$

in the SI, which matches Langevin simulations (*Figure 5d*). Since $\tau_{relax} \sim T_{day}$ to have distinct clock states through the day (*Figure 3*), we find $\sigma_{cycle}^{2,int} \geq \sigma_{point}^{2,int}$.

In summary, in both cases, population variance is reduced by the geometric 'curvature' of the dynamics, that is, convergence of nearby trajectories. Point attractor trajectories experience a constant curvature of $1/\tau_{relax}$. But limit cycle clocks experience such 'curved' off-attractor dynamics only at dawn and dusk, which is offset by dephasing (*Mihalcescu et al., 2004*; *Gonze et al., 2002*) during long periods of zero curvature on the limit cycle (day/night). Hence limit cycles underperform point attractors under high internal noise.

## Combination of external and internal noise

We now subject the clock systems to both internal and external noise at the same time. We find results (see *Figure 6a*) that parallel those for explicit molecular models of biochemical oscillators studied in *Figure 2*. Large-$R/L$ limit cycles outperform other clocks in filtering out external noise when internal noise is low, but their precision degrades more rapidly than other clocks as internal noise $\epsilon_{int}^2 \sim 1/N$ is increased. Point attractors have poor precision with only external noise but do not significantly degrade with internal noise and outperform all other clocks at high internal noise. At comparable strengths of internal and external noise, limit cycles with an intermediate value of $R/L$ are most precise. In the SI, we show that the optimal geometry is set by the ratio of internal and external noise strength,

$$(L/R)_{optimal} = \frac{\epsilon_{int}}{\epsilon_{ext}}. \tag{3}$$

In the SI, also we show that, under certain simplifying assumptions, *Equations 1 and 2* can be combined to give an explicit trade-off relationship,

$$\sigma_{int}^2 \sigma_{ext}^2 \sim Q \tag{4}$$

where $Q = \epsilon_{int}^2 \epsilon_{ext}^2$ and where $\sigma_{int}^2$ is the population angular variance of the clock state due to internal

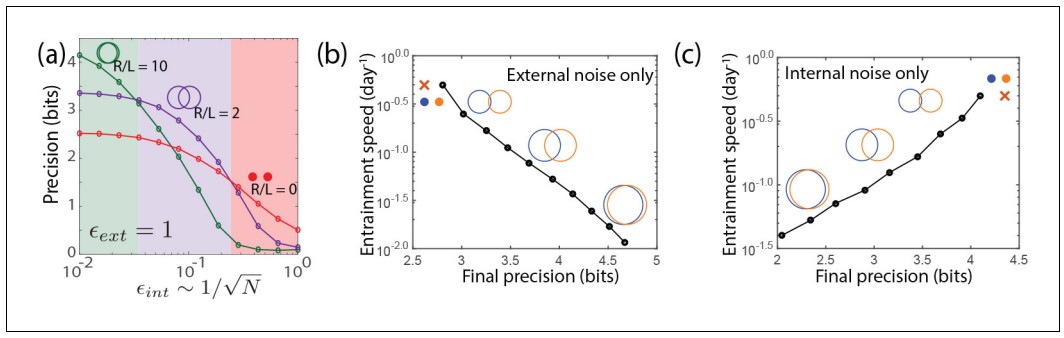

**Figure 6.** Large-$R/L$ limit cycle attractors, which correspond to large amplitude free running clocks, outperform all other oscillators in projecting out external noise but are least robust to internal noise. (a) Point attractors and smaller $R/L$ limit cycles (red and purple curves) show low precision (i.e., low mutual information) but do not degrade as much as large-$R/L$ limit cycles with increasing internal noise $\epsilon_{int}$. Thus this simple dynamical systems model of clocks reproduces and explains the trade-off seen in the complex biochemical clocks shown in *Figures 1* and *2*. (b,c) Speed-precision trade-off. (b) With external noise alone, the most precise clocks (i.e., large $R/L$ limit cycles) average over longer signal history and are thus the slowest to entrain, that is, slow to transform a population with uniform phase distribution to the steady state distribution. (c) However, with internal noise alone, there is no trade-off between speed and precision; faster entraining clocks (i.e., point attractors) are more accurate since slow clocks are exposed to more internal noise.
DOI: https://doi.org/10.7554/eLife.37624.008

noise when driven by a noiseless external signal and $\sigma_{ext}^2$ is the population angular variance in the absence of internal noise due to amplitude fluctuations of the external signal. Note that angular variance is a better indicator than variance because we want to know how well the system can tell time.

*Equation 4* makes our trade-off explicit and also clarifies which parameters are varied and which parameters are held fixed in this trade-off. As long as $Q$ is held fixed, we allow all other parameters to vary – for example, the overall strength of the external drive $L$, the size of the cycle $R$, and as discussed in the SI, all other parameters characterizing the normal form of limit cycles near a Hopf bifurcation.

However, in holding $Q$ fixed, our trade-off does assume that the strength of the external fluctuations $\epsilon_{ext}$ – that is, the fractional size of amplitude fluctuations in the external signal – is held fixed. Similarly, we hold $\epsilon_{int}^2$, the phase diffusion constant, fixed – that is, we are comparing clocks that would show the same population phase variance (in units of radians) over the same time in undriven conditions. See Appendix 3 for alternative comparisons and other details.

## Speed-precision trade-off

Another measure of clock quality is the entrainment speed, that is, the time taken to reach steady state population variance, starting from a population uniformly distributed in clock phase. In *Figure 6b*, we see that with external noise only, the most precise clocks (i.e., small-$L/R$ limit cycles) are the slowest to entrain because they retain a longer history of the external signal, allowing them to average out external noise better.

But strikingly, such a speed-precision trade-off is absent if internal noise is high. In *Figure 6c*, only internal noise is present and the external signal has no fluctuations. We see that clocks most robust to internal noise are also the fastest to entrain. Intuitively, the phase of slow entraining clocks is affected by the cumulative effect of internal fluctuations over a longer period of time. With both external and internal noise present, clocks with intermediate entraining speed - that is, intermediate $(L/R)_{optimal} = \epsilon_{int}/\epsilon_{ext}$ - will have the highest precision.

## Discussion

Free running circadian clocks are a remarkable result of evolution in a changing but predictable environment and are thought to provide numerous benefits (*Troein et al., 2009*). Here, we showed that the limit cycle attractor underlying such a clock is able to effectively project out weather-related amplitude changes that are perpendicular to the flat direction of the attractor. Similar roles for the flat direction of continuous attractors in projecting out external (or input) fluctuations have been explored in neuroscience (*Burak and Fiete, 2012*); *Seung (1996)*, for example, for head and eye motor control and spatial navigation. However, we also found that the same flat direction becomes a vulnerability with internal fluctuations since such fluctuations cannot be restricted to be perpendicular to the attractor.

We confirmed the trade-off between resistance to external and internal noise in diverse models of biochemical clocks and oscillators, using parameters inferred from experimental data by the original publications (*Leloup et al., 1999*; *Schmal et al., 2014*; *Locke et al., 2005*; *Leloup and Goldbeter, 2003*; *Goldbeter, 1991*; *Goodwin, 1965*; *Gonze and Abou-Jaoudé, 2013*; *Kondepudi and Prigogine, 2014*; *Elowitz and Leibler, 2000*; *Buşe et al., 2009*). The trade-off in each of these models can be given explanations that are specific to those models; for example, one can identify specific bottlenecks for external and internal noise in these models (*Cheong et al., 2011*). However, we have provided an alternative kind of analysis based on the geometry of the dynamical systems involved. Such an explanation misses aspects specific to each clock - for example, how specific biologically tuneable parameters in each model affect internal and external noise resistance. However, the dynamical systems picture has the advantage in that it identifies the common origin of the trade-off across these systems. Such a dynamical systems picture has been fruitful in making other general but falsifiable predictions in biology (*Gan and O'Shea, 2017*; *Leypunskiy et al., 2017*; *Corson and Siggia, 2017*), going back to Winfree's phase singularity (*Winfree, 2001*).

Our dynamical systems theory shows that the critical parameter for noise resistance is the strength of the external driving relative to the amplitude of free running oscillations, captured by the geometric ratio $L/R$ in our analysis. Weak driving provides resistance to external noise while strong

driving provides resistance to internal noise. While our dynamical systems theory involve planar circular limit cycles, the models in *Figure 2* have complex non-planar non-circular limit cycles and yet reproduce our trade-off. Finally, while the internal noise discussed here is set by finite copy number, this dependence is not essential to the results here. Any source of disturbance (e.g., bursty transcription) that perturbs the phase of the oscillator in constant light conditions is equivalent to internal noise. Similarly, external noise can involve any kind of fluctuation (e.g., multiplicative fluctuations, phase fluctuations) of the external signal that does not result in a persistent phase shift of the external signal itself.

Our work suggests that the damped oscillators are not merely poor cousins of the remarkable free running oscillators found for example, in *S. elongatus*. At the low protein copy numbers such as those found in *P. marinus*, damped clocks keep time more reliably than free running clocks. (Low copy number has been linked to a trend towards reduced genome size and cell size in *P. marinus* [*Bryant, 2003*].) In addition to the noisy internal environment of *P. marinus*, the external environment might also play a role in selecting a damped clock; *P. marinus* is typically found in the open ocean, where the external environment may be more regular than the fresh water habitat of *S. elongatus*. In addition to *P. marinus*, damped oscillators are found elsewhere in biology, often in specific physiological conditions (*Saunders, 2002*; *Kidd et al., 2015*). Understanding the benefits and drawbacks of such damped oscillators in different conditions is critical since such oscillations are easily overlooked experimentally, in comparison to free running oscillations.

While numerous upstream and downstream considerations can modify (*Rand et al., 2004*; *Domijan and Rand, 2011*) the ultimate biological impact of fluctuations, we find that the core oscillator's geometry in itself can continuously trade off protection against external fluctuations for protection against internal fluctuations in the diverse range of models studied here.

*Note added in proofs:* The study of Monti et al. (2018, in press) independently arrived at the conclusion that free running clocks based on limit-cycles are more robust to external noise. Experiments in Chew et al. (2018, in press) suggest that the free running clock in *S. elongatus* is less robust to internal noise than the hourglass clock in *P. marinus*.

## Materials and methods

We incorporated most of our methods in Results and Discussion. For details beyond those presented in Results, please see Appendices. Code to simulate the systems can be found at https://github.com/WeerapatP/elife_tradeoff_clocks (Pittayakanchit, 2018; copy archived at https://github.com/elifesciences-publications/elife_tradeoff_clocks).

## Acknowledgments

We thank Aaron Dinner, John Hopfield, Eugene Leypunskiy, Charles Matthews, Brian Moths, Thomas Witten, and the Rust and Murugan labs for fruitful discussions.

## Additional information

### Funding

| Funder | Author |
| --- | --- |
| Simons Foundation | Arvind Murugan |

The funders had no role in study design, data collection and interpretation, or the decision to submit the work for publication.

### Author contributions

Weerapat Pittayakanchit, Zhiyue Lu, Conceptualization, Software, Formal analysis, Visualization, Methodology, Writing—original draft, Writing—review and editing; Justin Chew, Conceptualization, Software, Validation, Investigation, Writing—review and editing; Michael J Rust, Conceptualization, Investigation, Writing—review and editing; Arvind Murugan, Conceptualization, Formal analysis,

Supervision, Funding acquisition, Visualization, Methodology, Writing—original draft, Project administration, Writing—review and editing

## Author ORCIDs
Weerapat Pittayakanchit ![ORCID] http://orcid.org/0000-0001-7940-3184
Zhiyue Lu ![ORCID] https://orcid.org/0000-0002-0216-4346
Justin Chew ![ORCID] http://orcid.org/0000-0003-4749-547X
Michael J Rust ![ORCID] http://orcid.org/0000-0002-7207-4020
Arvind Murugan ![ORCID] http://orcid.org/0000-0001-5464-917X

## Decision letter and Author response
Decision letter https://doi.org/10.7554/eLife.37624.021
Author response https://doi.org/10.7554/eLife.37624.022

## Additional files

### Supplementary files
• Transparent reporting form
DOI: https://doi.org/10.7554/eLife.37624.009

### Data availability

The code and data used in the simulations are available via GitHub at https://github.com/WeerapatP/elife_tradeoff_clocks

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

## Appendix 1

DOI: https://doi.org/10.7554/eLife.37624.010

## Trade-off in Kai-based clocks

We demonstrate our trade-off using Gillespie simulations of an explicit biomolecular KaiABC model of the post-translational clocks in *S. elongatus* and *P. marinus*.

## *S. elongatus* clock - hexamers with collective KaiA feedback

The *S. elongatus* clock has been well-characterized experimentally (*Bryant, 2003*; *Gutu et al., 2013*; *Dufresne et al., 2003*; *Kitayama et al., 2003* - see *Appendix 1—figure 1a*). The clock is fundamentally based on the ordered phosphorylation and dephosphorylation of KaiC (*Rust et al., 2007*). Phosphorylation of KaiC is KaiA-dependent which allows for feedback that enables collective coherent oscillations in a cell. After complete phosphorylation of KaiA-C complexes (usually by the end of the day), KaiC forms a KaiB-C complex which then dephosphorylates in an ordered manner. Crucially, the KaiB-C complex also sequesters KaiA in a KaiABC complex, reducing the pool of available KaiA for phosphorylation of other KaiC hexamers. This negative feedback enables coherent oscillations of the population of KaiC molecules in a single cell (*Rust et al., 2007*).

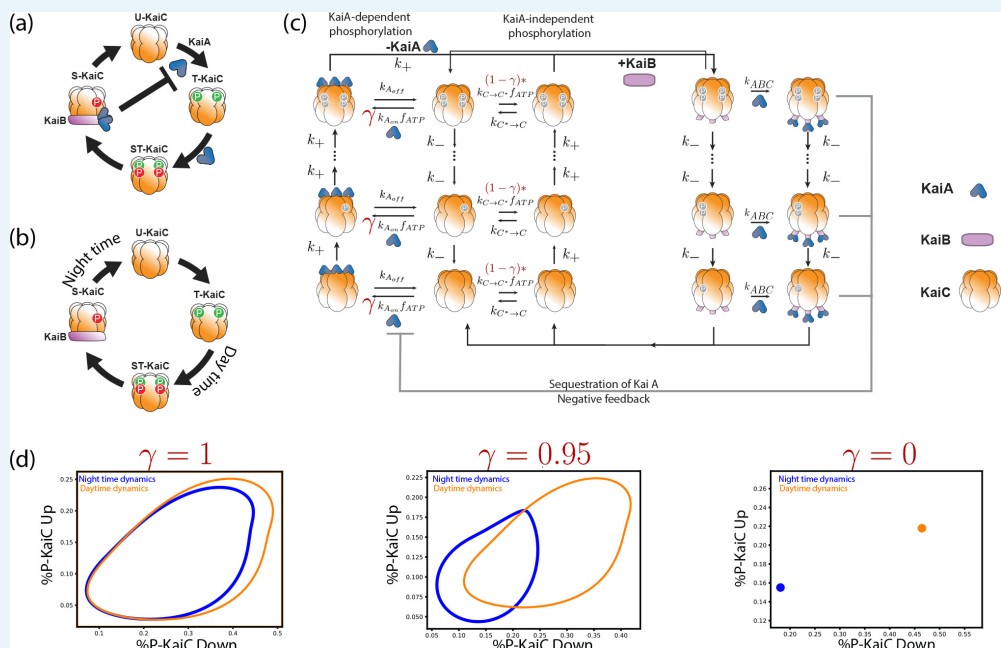

**Appendix 1—figure 1.** Explicit biochemical KaiABC model simulated using the Gillespie algorithm. (**a**) The experimentally well-characterized clock in *S. elongatus* consists of a negative feedback-enabled self-sustained oscillator. KaiBC complexes sequester KaiA, preventing runaway KaiC molecules from going through the cycle independently. (**b**) The genome of *P. marinus* lacks *kaiA*. We assume a minimal model consistent with known facts (*Rust et al., 2007*) about this clock; KaiC phosphorylation proceeds without KaiA and hence different KaiC hexamers can proceed independently through the cycle. (**c**) We combine both clocks in one model with an interpolating parameter $\gamma$ that selects between an *S. elongatus*-like KaiA-dependent pathway and an *P. marinus*-like KaiA-independent pathway. All reactions shown are assumed to be first order mass-action kinetics. We simulate such a system at different overall copy numbers $N$ using the Gillespie algorithm. (**d**) We find limit cycles for $\gamma > 0.9$. The resulting limit cycles for $\gamma = 1, 0.95$ violate the simplifying assumptions used in our dynamical

systems (e.g., non-circular cycles of different size); and yet our results are qualitatively validated by this model (**Figure 1d** from the main text).

DOI: https://doi.org/10.7554/eLife.37624.011

### *P. marinus* model - independent hexamers

*P. marinus* lacks the *kaiA* gene but possesses and expresses *kaiB* and *kaiC*. While the details of the protein clock dynamics are not as fully known as with *S. elongatus*, gene expression shows cycling in cycling conditions but decays in constant conditions (**Holtzendorff et al., 2008**). A conservative model, consistent with all these known facts about *P. marinus*, is shown in **Appendix 1—figure 1b**; without KaiA feedback, different hexamer units phosphorylate independently and settle to a hyperphosphorylated state at the end of the day. At night, they dephosphorylate along a distinct pathway (homologous to that used by *S. elongatus* but without KaiA) and reach a hypophosphorylated state by dawn.

### Hybrid model

We created the following hybrid model that includes *S. elongatus* and *P. marinus* models as different limits. In our model, shown in **Appendix 1—figure 1c**, KaiC has a KaiA-dependent phosphorylation pathway, much like in *S. elongatus*, that is used during the day and driven forward by ATP.

But to also include *P. marinus*-like behavior in the model, we allow for a second parallel phosphorylation pathway for KaiC that is independent of KaiA. The relative access of these two pathways is controlled by a parameter $\gamma$. When $\gamma = 1$, only the *S. elongatus*-like KaiA dependent pathway is accessible. When $\gamma = 0$, only the *P. marinus*-like KaiA independent pathway is accessible. Collectively, we call these states along these phosphorylation pathways, the UP states of KaiC - phosphorylation are going UP along these pathways which are usually used during the day.

After maximum phosphorylation (usually at dusk), KaiA unbinds (if present) and a KaiB-based dephosphorylation pathway takes over (common to both systems). We call these states the DOWN states of KaiC.

Critically, KaiA is assumed to be sequestered through the formation of KaiABC complexes during this dephosphorylation stage. In *S. elongatus*, reduced KaiA availability prevents other KaiC hexamers from proceeding independently through the UP stage while most of the population is in the DOWN state. Such negative feedback is critical in maintaining free-running limit cycle oscillations in *S. elongatus*.

However, as $\gamma \to 0$, the KaiA-independent pathway is more active and thus the system effectively has no feedback. In fact, we find that at about $\gamma \approx 0.82$, sustained oscillations disappear (for kinetic parameters used here and reported below). Hence we chose $\gamma = 1, 0.95, 0$ as representative of two limit cycle-based free running clocks and one point-attractor based damped clock respectively. In this way, we can view the clock dynamics of *S. elongatus* and *P. marinus* can be viewed as being on either side of the Hopf bifurcation that occurs at $\gamma \approx 0.82$.

### Gillespie simulations

We ran explicit Gillespie simulations corresponding to the deterministic equations above at different overall copy number $N$ with fixed stoichiometric ratios of the molecules KaiA,B,C.

We simulated external input noise by varying the ATP levels during the day. External noise in these simulations were implemented by changing ATP levels in the following way: we fluctuated the ATP levels $f_{ATP} = ATP/(ATP + ADP)$ during the day between the $f_{ATP}^{day}$ and $f_{ATP}^{night} + (f_{ATP}^{day} - f_{ATP}^{night})/3$, where $f_{ATP}^{day}, f_{ATP}^{night}$ are the ATP values during a cloudless day and night respectively. We used different day and night ATP levels for different $\gamma$ that ensure that the limit cycles had periods comparable to 24 hours. For $\gamma = 1$, we used ATP/ADP ratios of $f_{ATP}^{day} = 0.55, f_{ATP}^{night} = 0.45$. For $\gamma = 0.95$, we used $f_{ATP}^{day} = 0.57, f_{ATP}^{night} = 0.17$ and for $\gamma = 0$, $f_{ATP}^{day} = 0.8, f_{ATP}^{night} = 0.2$. The corresponding limit cycles and point attractors are shown in **Appendix 1—figure 1d**.

We used the following kinetic parameters in all simulations: $dt = 0.01\ hr$, $k_+ = k_- = 2m \cdot 0.04932\ hr^{-1}$, $k_{Aon} = 0.2466\ \mu M^{-1} hr^{-1}$, $k_{Aoff} = 0.02466\ hr^{-1}$, $k_{C \to C*} = 0.2466\ hr^{-1}$, $k_{ABC} = 123.30\ hr^{-1}$, $m = 18$. We set up Kai C and Kai A in a $1:1$ stoichiometric ratio, each present at a copy number $N$ where $N$ was varied systematically. These rates are consistent with those measured in (*Qin et al., 2010*; *Snijder et al., 2017*; *Hayashi et al., 2004*).

Much like with Langevin simulations of dynamical systems performed in this paper, we run the Gillespie simulation until equilibration of the population. However, the system appears to reach the equilibrium state much faster (only over five light-dark cycles of 12 hr: 12 hr). We extracted one day of such a trajectory on day six and repeated the simulation 100–400 times. We repeat 400 times when the copy number is low (<1200) since the spread will be big and we found that the probability distribution is not smooth. We run only 100 times for the high copy number (>1200). Pooling together these trajectories, we computed the mutual information between clock state (i.e., $(u, d)$ where $u$ is the net phosphorylation state of KaiC in the up-pathways and $d$ is the net phosphorylation state of KaiC in the KaiB-bound 'down' pathways in *Appendix 1—figure 1c* ) and time of day. The $(u, d)$ space was binned using bins of fixed size of dimension $(0.05, 0.05)$ while the 24 hr time-of-day was binned with bins of size $0.5$ hrs.

## Phase portrait

With these choices, we see in *Appendix 1—figure 1d* that this model has limit cycles of different position during the day and night. The corresponding experimental data, reproduced from *Leypunskiy et al. (2017)*, are presented in the main paper.

## Appendix 2

DOI: https://doi.org/10.7554/eLife.37624.012

## Other oscillators

Here, we study the effect of internal and external noise on a diverse array of biochemical oscillator models from the literature in the parameter regimes described in the original publications. We confirm the same trade-off described in the paper in these models; a summary of our results is presented in the main paper.

In all of the models described below, we set all parameters to values used in the original or cited papers with only two exceptions: (a) the parameter identified as coupling to external signals in these publications is varied over time as a square wave with amplitude fluctuations added, (b) the parameter designated by the relevant original publication as controlling the distance from the Hopf bifurcation was used to simulate a point attractor-based 'hourglass' oscillator (red lines in *Figure 2* of the main paper) and limit cycles of different free running oscillation amplitude (green and purple lines in *Figure 2* of the main paper). This latter parameter roughly corresponds to $R$, the size of limit cycle, while the amplitude of square wave coupled to the former parameter corresponds to $L$, the separation of the limit cycles, in our dynamical systems theory, that is, the separation of the 'day' and 'night' limit cycles. (In several papers, these two are the same parameter, in which case the day-night difference reflects $L$ while the mean value reflects $R$.) Finally, we add Langevin noise to the equations to simulate internal noise; when available, we followed the finite volume prescription for rates in these papers to set the size of Langevin noise for each reaction.

We keep the strength of external noise $\epsilon_{ext}$, defined as the noise-to-signal ratio of the amplitude fluctuations in the external signal, fixed. We varied internal noise $\epsilon_{int}$ along the $x$ axis of plots in *Figure 2* of the main paper. Here, $\epsilon_{int}$ is defined as the *phase* diffusion constant of a clock in undriven conditions (see how we define internal noise in the section on Neurospora and Drosophilia below); this normalization, which depends on the Hopf bifurcation parameter in (b) above, allows us to make a fair comparison between different clocks since they develop the same phase variance over the same time in undriven conditions.

As seen in *Figure 2* of the main paper, these diverse models agree with the trends found in our analysis of dynamical systems and with simulations of the KaiABC system, showing that our results are not tied to any particular molecular model.

### *Neurospora* and *Drosophila* circadian clocks by Goldbeter

The circadian clock in *Neurospora* has been modeled (*Leloup et al., 1999*) as arising from interactions between mRNA ($M$) and a protein that can shuttle in and out of a nucleus ($P_N, P_C$). The equations used in *Gonze and Goldbeter (2006)* to model this are,

$$\frac{dM}{dt} = v_s \Omega \frac{(K_I \Omega)^n}{(K_I \Omega)^n + P_N^n} + v_m \Omega \frac{M}{K_m \Omega + M}$$

$$\frac{dP_C}{dt} = k_s M - v_d \Omega \frac{P_C}{K_d \Omega + P_C} - k_1 P_C + k_2 P_N$$

$$\frac{dP_N}{dt} = k_1 P_C - k_2 P_N \tag{5}$$

where $v_s$ is an mRNA transcription rate that is modulated by external signals (*Leloup et al., 1999*; *Gonze and Goldbeter, 2006*), and $\Omega$ is the volume of the system which in turn determines the strength of stochastic noise. (A model with very similar equations has also been suggested as a model of the *Drosophila* circadian clock [*Leloup et al., 1999*].)

We use the same parameters used in the Ref. (*Leloup et al., 1999*; *Gonze and Goldbeter, 2006*): $K_I$ = 1 nM, $n$ = 4, $v_m$ = 0.505 nM/h, $K_m$ = 0.5 nM, $k_s$ = 0.5 1/h, $v_d$ = 1.4 nM/h, $K_d$ = 0.13 nM, $k_1$ = 0.5 nM/h, $k_2$ = 0.6 nM/h, and assume the volume $\Omega$ dependence of these parameters

to be exactly as used in *Gonze and Goldbeter (2006)*. We add internal stochasticity by adding Langevin noise with a diffusion matrix (*Gonze and Goldbeter, 2006*):

$$\frac{dX}{dt} = \mu(x,t) + \Sigma(x,t)\eta(0,1) \tag{6}$$

where $\mu(x,t)$ is the RHS of *Equation 5*, $\eta(0,1)$ is a vector whose entries are independent standardized Gaussian noise (mean 0, variance 1), and

$$\Sigma(x,t) = \begin{bmatrix} \sqrt{A} & \sqrt{B} & 0 & 0 & 0 & 0 & 0 \\ 0 & 0 & \sqrt{k_s M} & \sqrt{k_1 P_C} & \sqrt{k_2 P_N} & 0 & 0 \\ 0 & 0 & 0 & 0 & 0 & \sqrt{k_1 P_C} & \sqrt{k_2 P_N} \end{bmatrix} \tag{7}$$

where $A = v_s \Omega \frac{(K_I \Omega)^n}{(K_I \Omega)^n + P_N^n}$ and $B = v_m \Omega \frac{M}{K_m \Omega + M}$. This is how internal noise get added into other oscillators models as well. However, for the system of equations that use concentration instead of the number of molecules, the equation has to be modified to $\frac{dX}{dt} = \mu(x,t) + \frac{1}{\sqrt{\Omega}}\Sigma(x,t)\eta(0,1)$.

As in *Leloup et al. (1999)* and *Gonze and Goldbeter (2006)*, we take $v_s$ to be modulated by the external signal (light). As shown in *Leloup et al. (1999)* and *Gonze and Goldbeter (2006)*, a Hopf bifurcation occurs at $v_s = 0.57$ nM/h. Hence, in *Figure 2* from the main text, we used $v_s^{Day} = 0.55$ nM/h, $v_s^{Night} = 0.05$ nM/h for the point attractor (red). For the two limit cycles, we used $v_s^{Day} = 0.9$ nM/h, $v_s^{Night} = 0.6$ nM/h (green), and $v_s^{Day} = 0.705$ nM/h, $v_s^{Night} = 0.695$ nM/h (purple). The driving period is 18 hr, similar to the driving period of the system at $v_1 = 0.7$ nM/h

## *Arabidopsis* circadian clock by Millar et al

A model of the circadian clock in *Arabidopsis thaliana* was introduced in *Locke et al. (2005)*. While many biologically important features have been added in the years since then, the original model was based on a single negative feedback loop and involves two transcription factors (LHY and CCA1) that inhibit their activator $TOC1$. A reduced model with the same phenomenology was presented in *Schmal et al. (2014)*, in which LHY and CCA1 are combined into one variable, representing their mRNA and protein levels by $M_L(t)$ and $P_L(t)$ respectively. Denoting the mRNA and protein levels of $TOC1$ by $M_T(t)$ and $P_T(t)$, *Schmal et al. (2014)* present a reduced version of the model in *Locke et al. (2005)* as:

$$\frac{dM_L}{dt} = L(t) + v_1 \frac{P_T^a}{g_1^a + P_T^a} - m_1 \frac{M_L}{k_1 + M_L}$$

$$\frac{dP_L}{dt} = p_1 M_L - m_2 \frac{P_L}{k_2 + P_L}$$

$$\frac{dM_T}{dt} = v_2 \frac{g_2^b}{g_2^b + P_L^b} - m_3 \frac{M_T}{k_3 + M_T}$$

$$\frac{dP_N}{dt} = p_2 M_T - m_4 \frac{P_T}{k_4 + P_T} \tag{8}$$

where $L(t)$ is a light input function, and other parameters except the variables specified on the left hand sides are constant.

To simulate this system, we use the parameters used in *Schmal et al. (2014)*: a = 2, b = 2, $g_1 = 0.5$ nM, $g_2 = 0.1$ nM, $m_1 = 0.4$ nM/h, $m_2 = 0.6$ nM/h, $m_3 = 0.6$ nM/h, $m_4 = 0.3$ nM/h, $k_1 = 1$ nM, $k_2 = 0.5$ nM, $k_3 = 1$ nM, $k_3 = 1$ nM, $p_1 = 0.5$ 1/h, $p_2 = 0.3$ 1/h, $v_2 = 0.6$ nM/h. With other parameters fixed, the system undergoes Hopf bifurcation at $v_1 = 0.194$ nM/h We use $v_1 = 0.26$ nM/h for limit cycles and $v_1 = 0.05$ nM/h for point attractor. $L(t)$ is a light input function. For the two limit cycles in *Figure 2* in the main text, we set $L^{Day} = 0.05$ nM/h and $L^{Night} = 0$ (green data) and we set $L^{Day} = 0.01$ nM/h and $L^{Night} = 0$ (purple data). For point attractor, we set $L^{Day}$

= 0.2 nM/h and $L^{Night}$ = 0 (red data). The period of the driving signal is 24 hr, which is around the natural period of the system when $v_1$ = 0.26 nM/h and $L$ = 0.

## B.3 Mammalian *Per-Cry* circadian clock by Leloup et al

The circadian clock in mammalian cells was modeled in detail by *Leloup and Goldbeter (2003)*, using 19 equations representing the interactions between *Per*, *Cry* and other genes. We simulate this entire system explicitly with the parameter values specified in the original publication (*Leloup and Goldbeter, 2003*). To introduce Langevin noise, we use a simplified diagonal diffusion matrix with entry $\sqrt{DX_i}$ for species $X_i$. We do not reproduce these 19 equations or parameter values used from *Leloup and Goldbeter 2003*) here in interest of space; the only modification we made is to introduce Langevin noise to each of the 19 equations.

*Leloup and Goldbeter (2003)* identified parameter $v_{sP}$ (a transcriptional rate) to be the light input function. We use $v_{sP}^{Day}$ = 1.09 nM/h and $v_{sP}^{Night}$ = 1.07 nM/h for the purple limit cycle data in *Figure 2*, $v_{sP}^{Day}$ = 1.15 nM/h and $v_{sP}^{Night}$ = 1.07 nM/h for the green limit cycle data. For the point attractor data (red), we set $v_{sP}^{Day}$ = 1.5 nM/h and $v_{sP}^{Night}$ = 0. In addition, *Leloup and Goldbeter (2003)* identified parameters $KAC, vmB$ as controlling the distance from the Hopf bifurcation. For the point attractor, we used KAC = 0.4 nM, and vmB = 0.9 nM/h (also used in *Leloup and Goldbeter, 2003*). For the limit cycles, we used KAC = 0.6 nM, and vmB = 0.8 nM/h, which lies on the other side of the Hopf bifurcation. The period of the input signal is at 21.5 hr, which is around the natural period of the system when $v_{sP}$ = 1.07 nM/h.

## B.4 cdc2-cyclin cell cycle by Goldbeter

A classic model of the cell cycle was proposed by *Goldbeter (1991)*. While many additional details have been added on since then, the model captures the essential mechanism behind the self-sustained nature of cell cycles.

$$\frac{dC}{dt} = v_i\Omega - k_dC - v_dX\Omega\frac{C}{K_d\Omega + C}$$

$$\frac{dM}{dt} = v_1\frac{C}{K_C\Omega + C}\frac{1 - M}{K_1 + (1 - M)} - V_2\frac{M}{K_2 + M}$$

$$\frac{dX}{dt} = v_3M\frac{(1 - X)}{K_3 + (1 - X)} - V_4\frac{X}{K_4 + X} \tag{9}$$

where $\Omega$ is the size of the system and other parameters are constants. The three variables are the cyclin concentration $C$, the fraction of active *cdc2* kinase $M$, and the fraction of active cyclin protease $X$. For $C$, the internal noise is proportional to the square root of the rates, but for $M$ and $X$, it is proportional to the square root of the rates divided by $\Omega$ because they are fractions and not concentrations (following the prescription in *Gonze and Goldbeter, 2006* for a similar model). Parameter values were taken from *Goldbeter (1991)*: $K_i = 0.1$ ($i$=1–4), $V_{M1} = 0.5 \sim \min^{-1}$, $V_2 = 0.167 \sim \min^{-1}$, $V_{M3} = 0.2 \sim \min^{-1}$, $V_4 = 0.1 \sim \min^{-1}$, $v_d = 0.1 \sim \mu M\min^{-1}$, $K_C = 0.3\mu M$, $K_d = 0.02 \sim \mu M$, $k_d = 3.33 \times 10^{-3} \sim \min^{-1}$.

Goldbeter (*Goldbeter, 1991*) suggested that $v_i$ is modulated by external signals. So, we use $v_i^{Day}$ = 0.0106 $\sim \mu M\min^{-1}$ and $v_i^{Night}$ = 0.0105 $\sim \mu M\min^{-1}$ for small L/R limit cycle, $v_i^{Day}$ = 0.0111 $\sim \mu M\min^{-1}$ and $v_i^{Night}$ = 0.0105 $\sim \mu M\min^{-1}$ for large L/R limit cycle, and $v_i^{Day}$ = 0.009 $\sim \mu M\min^{-1}$ and $v_i^{Night}$ = 0 for point attractor. The bifurcation from point attractor to limit cycle happen around $v_i$ = 0.01 $\sim \mu M\min^{-1}$. The period of the driving signal is 35 min.

## B.5 Goodwin oscillator

One of the earliest models of biochemical oscillators was proposed by *Goodwin (1965)* (later corrected). We use the simplest widely-studied version of such a Goodwin oscillator (*Gonze and Abou-Jaoudé, 2013*; *Woller et al., 2014*),

$$\frac{dX}{dt} = \frac{\alpha(t)}{1 + Z^n} - X$$

$$\frac{dY}{dt} = X - Y$$

$$\frac{dZ}{dt} = Y - Z \tag{10}$$

When $n = 9$, the limit cycles disappear at a Hopf bifurcation found at $\alpha \approx 7$. As is commonly done (*Woller et al., 2014*), we couple the external signal to the bifurcation parameter $\alpha(t)$. We use $\alpha^{Day} = 120, \alpha^{Night} = 80$ for the green limit cycle in *Figure 2c* of the main paper, $\alpha^{Day} = 108, \alpha^{Night} = 92$ for purple limit cycle data, and $\alpha^{Day} = 2.5, \alpha^{Night} = 1$ for the red point attractor data. The input signal has a period of 4, which is roughly the natural period of the limit cycle at $\alpha = 100$ are taken to the output of the clock for computing MI.

## B.6 Repressilator

The repressilator is a model of an early synthetic biology system (*Elowitz and Leibler, 2000*) that demonstrated oscillations in a synthetically wired gene regulatory circuit. While resembling the Goodwin oscillator in topology, the network has the total non-linearity spread equally amongst all three reactions, lowering the requisite Hill coefficient of any one reaction to a biochemically realistic $n = 3$. Repressilator circuits are not usually driven by an external signal, except in a few theoretical analyses (e.g., *Russo et al., 2010*; *Schultz et al., 2013*). We use the simplest version of these, with the driving signal modulating the transcription rate of only one of the proteins

$$\frac{dX}{dt} = \frac{\alpha * (1 + s(t))}{1 + Y^n} - X$$

$$\frac{dY}{dt} = \frac{\alpha}{1 + Z^n} - Y$$

$$\frac{dZ}{dt} = \frac{\alpha}{1 + X^n} - Z \tag{11}$$

where $\alpha$ is the bifurcation parameter and $s(t)$ is the variable coupled to the input signal. In a non-driven repressilator, $s(t) = 0$. When $n = 3$, the Hopf bifurcation occurs at $\alpha \approx 2.5$, so for limit cycles, we use $\alpha = 5.2$ and for point attractor we use $\alpha = 1.9$.

We use $s^{Day} = 0.7/5.2, s^{Night} = -0.7/5.2$ for the green limit cycle in *Figure 2g* in the main paper, $s^{Day} = 4.8/5.2, s^{Night} = -1.7/5.2$ for the purple limit cycle data, and $s^{Day} = 0.5/1.9, s^{Night} = -1.9/1.9$ for the point attractor (red). The input signal has a period of 4, which is roughly the natural period of the limit cycle at $\alpha = 5.2$ and $s(t) = 0$ are taken to the output of the clock for computing $MI$.

## B.7 Brusselator

The Brusselator is a model of autocatalytic reactions that show limit cycle oscillations. This model has been extensively studied over the years; while the explicit biochemical reactions can be found in *Kondepudi and Prigogine, 2014*), these reactions are modeled by the ODEs:

$$\frac{dX}{dt} = 1 - (1 + b(t))X + X^2 Y$$

$$\frac{dY}{dt} = b(t)X - X^2 Y \tag{12}$$

where $b$ has been identified as a bifurcation parameter (**Kondepudi and Prigogine, 2014**). Most studies do not consider driven Brusselator models; we follow the driving prescriptions of the Goodwin model and couple the external light to the bifurcation parameter, converting the constant $b$ into $b(t)$. The bifurcation point is at $b = 2$. For $b < 2$, we have a point attractor and for $b > 2$ we have a limit cycle. We use $b^{Day} = 2.25$ and $b^{Night} = 2.2$ for the purple limit cycle data in **Figure 2f** from main text. We use $b^{Day} = 2.8$ and $b^{Night} = 2.2$ for the green limit cycle data. Lastly, we use $b^{Day} = 1.8$ and $b^{Night} = 0.5$ for the point attractor (red). The signal has a period of 6.4 which is around the natural period of the system when $b = 2.2$.

## Appendix 3

DOI: https://doi.org/10.7554/eLife.37624.013

### Dynamical Systems

To complement our study of detailed biochemical implementations of such systems, we study two kinds of dynamical systems in this paper; limit cycles and point attractors. The minimal model of limit cycles and point attractors is given by the 'normal form' near a Hopf bifurcation:

$$\dot{r} = \alpha\left(r - \frac{r^3}{R^2}\right) \tag{13}$$

$$\dot{\theta} = \omega \tag{14}$$

For $\alpha > 0$, the above equation describes a circular limit cycle of radius $R$ and frequency $\omega$. This equation undergoes a Hopf bifurcation at $\alpha = 0$, where the limit cycle shrinks to zero and resulting in point attractor for $\alpha < 0$. The 'normal form' can be seen as the universal simple form – for example, circular limit cycles of radius $R$ – that any limit cycle and point attractor will reduce to in the neighborhood of a Hopf bifurcation. We add White noise in the Cartesian space representation of the Dynamical equations to represent the internal noise as follows:

$$\mathrm{d}x = \left(\alpha\left(1 - \frac{x^2+y^2}{R^2}\right)x - \omega y\right)\mathrm{d}t + \sqrt{2D}\,\tilde{~}\,\mathrm{d}W \tag{15}$$

$$\mathrm{d}y = \left(\alpha\left(1 - \frac{x^2+y^2}{R^2}\right)y + \omega x\right)\mathrm{d}t + \sqrt{2D}\,\tilde{~}\,\mathrm{d}W \tag{16}$$

where $D \sim R^2 \epsilon_{int}^2$ is the diffusion constant, $\mathrm{d}W$ is a Wiener process, and here we assume that the internal noise is a homogeneous white-noise in the 2-dimensional space. (Similar assumptions are made in reference *Potoyan and Wolynes, 2014*)

While we assume this simple form here as a minimal model, we do not assume that the oscillator is weakly driven. Instead, based on experimental observations of the Kai clock (*Leypunskiy et al., 2017*) and models of numerous other clocks (*Winfree, 2001*), we assume that the origin of the limit cycle or point attractor equations above moves by a finite amount $L$ as the external light signal switches between day and night values. In fact, we move the origin along the x-axis as a function of time as $(Ls(t), 0)$ where $s(t)$ is the external light signal, assumed to be of amplitude 1. Thus we are assuming a simple circular form of limit cycles and point attractors but do not restrict to weak driving. (In the limit of weak driving, that is, small $L/R$, our model can be shown to reduce to the universal Stuart-Landau model of weakly driven oscillators as a special case.)

In *Equation 14*, $\tau_{\mathrm{relax}} \sim \frac{1}{|\alpha|}$ is the relaxation time for perturbations away from the limit cycle or point attractor. For limit cycles, perturbations away from the limit cycle tend to decay fast relative to the period $2\pi/\omega$, typically on the order of hours (*Leypunskiy et al., 2017*).

In contrast, the point attractor in damped 'hourglass' clocks *P. marinus* needs to have relaxation $\tau_{\mathrm{relax}} \sim \frac{1}{|\alpha|} \sim 2\pi/\omega$ comparable to the period of the day-night cycle itself. As explained in the main paper, if relaxation were much faster, the clock state would decay to a fixed point before the end of the day (or night) and thus not show distinct states at distinct times of the day-night cycle.

### Simulations

For both limit cycles and point attractors, we simulate a population of clocks, each represented by a particle in the given dynamical system, subject to external and/or internal noise.

We use $\alpha = 5$ for limit cycle system and $\alpha = -5$ for point attractor system where $\omega = 2\pi$ in these units. For point attractors, we set $R = 1000L$, where $L$ is the separation of the day and night attractor. In such a limit, the point attractors are quadratic potentials with linear restoring forces since $\frac{r^3}{R^2}$ is small. The center of the cycle and point attractors during the day are assumed to be at $(-L, 0)$ and at $(0, 0)$ at night; or more generally at $(-Ls(t), 0)$ were $s(t)$ the light signal (assumed to be of amplitude 1).

To simulate external noise, we use a square wave signal $s(t)$ of amplitude 1 with amplitude fluctuations set by $\epsilon_{ext}$. As explained in Appendix 4 on external noise, we take the fluctuations in $s(t)$ to have a correlation time of $2.4$ hours.

To simulate internal noise, we add Langevin noise to **Equations 14** as described in Appendix 6 on Langevin noise. As explained in Appendix 6, our measure of internal noise $\epsilon_{int}^2$ is a measure of *phase* diffusion, independent of limit cycle size. In other words, $\epsilon_{int}^2$ is a measure of the population phase variance (i.e., variance in $\theta$) developed by limit cycles of any size in undriven conditions in a given period of time.

To interpolate between limit cycles and point attractors, we systematically change $L$ holding $R$ fixed. For limit cycle simulations, changing $L$ and $R$ are equivalent. To see this for external noise simulations, note that $L/R$ is the only dimensionless parameter. For internal noise, our definition of $\epsilon_{int}^2$ above as the phase diffusion in undriven conditions for limit cycles of any size, ensures that changing $L$ and $R$ are equivalent for limit cycles.

For point attractors, we set the separation $L$ be the diameter of the limit cycles simulated in the same plots to keep the size of the resolvable chemical spaces roughly comparable for limit cycles and point attractors. While this precise choice is arbitrary to some extent, note that the point attractor results for external noise do not depend on this parameter at all since $L$ is the only relevant length scale in external noise simulations. The separation $L$ does affect the clock precision with internal noise (of fixed absolute strength $\epsilon_{int}$) but the mutual information changes only logarithmically with $L$.

We evolve our dynamical system using the Euler method with time step $dt = 0.001$ days until the value of mutual information from one day to the next does not change by more than 2–3% - that is. the system has reached steady state. Reaching steady-state usually takes around 200 days, but if the ratio of $L/R$ is smaller than $0.1$, then we may need to run the simulation until day 500 to reach an equilibrium (See speed-error tradeoff in **Figure 6b and c** in the main text).

For limit cycles, we initialize the population of $10^4$ particles by uniformly distributing them along the perimeter of the night cycle. In the point attractor system, we initialize a population of $10^5$ at the night-time point attractor.

We use a larger population with point attractors since the particles tend to be distributed over a larger area of the dynamical system. Note that we bin the population by position to compute mutual information between position in the 2d state space and time. Doing so reliably requires a smooth distribution after binning. For limit cycles, the particles usually stay close to attractor and thus provide sufficient count in each bin. However, for the point attractor, the population is usually spread over the entire 2d area between the two point attractors. Therefore, we need $10^5$ particles to get an accurate value of mutual information of point attractor system.

## Optimal Dynamical system and trade-off

To find the optimal dynamical system geometry that operates with best accuracy when both internal noise and external noises are present, recall that we derived the following equations for strongly-driven limit cycles,

$$\sigma_{int}^2 \sim \frac{\epsilon_{int}^2 T}{s^2 - 1} \tag{17}$$

$$\sigma_{ext}^2 \sim \frac{\Delta\Phi^2}{s^4 - 1} \tag{18}$$

For a small $L/R$, we had found that $\Delta\Phi \sim \epsilon_{ext}L/R$ where $\epsilon_{ext}^2$ is a measure of the variance of light during the day. Further, we showed that $s^2 - 1 \sim L/R$ in this limit. Hence, in the small $L/R$ ('Stuart-Landau') regime, the above equations reduce to,

$$\sigma_{int}^2 \sim \epsilon_{int}^2 R/L \tag{19}$$

$$\sigma_{ext}^2 \sim \epsilon_{ext}^2 L/R \tag{20}$$

The population variance when both noises are present is approximately given by $\max(\epsilon_{int}^2 R/L, \epsilon_{ext}^2 L/R)$. This variance is minimized when the two terms are equal, giving

$$\left(\frac{L}{R}\right)_{optimal} \sim \frac{\epsilon_{int}}{\epsilon_{ext}},$$

which defines the optimal geometry of the dynamical system for given strength of internal and external noise.

In contrast, by taking the product of the equations above, we find the trade-off relationship,

$$\sigma_{ext}^2 \sigma_{int}^2 \sim Q \equiv \epsilon_{ext}^2 \epsilon_{int}^2 \tag{21}$$

The trade-off relationship above clarifies which parameters are held fixed and which ones are varied in our trade-off. If $Q$ is held fixed, this trade-off relationship holds under variations of all the parameters of the normal form of limit cycles (**Equation 14**). (While $L/R$ allows us to navigate the trade-off by increasing one of $\sigma_{ext}^2, \sigma_{int}^2$ and decreasing the other, other parameters such as the relaxation time leave both $\sigma_{ext}^2, \sigma_{int}^2$ relatively unaffected.)

Holding $Q$ fixed does involve holding the strength of external $\epsilon_{ext}$ and internal $\epsilon_{int}$ noise fixed. In all the models studied here, $\epsilon_{ext}$ is simply defined as the size of the amplitude fluctuations in the external signal relative to the amplitude of the external signal itself - that is, the noise-to-signal ratio of the external signal - with no reference to the clock dynamics. Changing $L/R$ and other parameters can strengthen or weaken the coupling of this noisy external signal to the clock but do not affect the signal-to-noise ratio of the external signal itself.

Analogously, the strength of internal noise $\epsilon_{int}^2$ is defined as the diffusion constant for the *phase* of a clock (e.g., in radians$^2$) in the absence of an external driving signal. As discussed in Appendix 3, this definition ensures that limit cycles of different sizes develop the same phase variance over the same time when subject to the same $\epsilon_{int}$.

For some purposes, it may make sense to hold the dimensionful diffusion constant $D_{int} = \epsilon_{int}^2 R^2$ fixed while making comparisons. In this case, in addition to the trade-off effect discussed in this paper, large limit cycles are given an additional robustness to internal noise, trivially by virtue of their size, since the diffusion constant $D_{int}$ in chemical space is held fixed (instead of the dimensionless diffusion constant $\epsilon_{int}^2$ for clock phase). In this case, it is insightful to re-write $Q = \epsilon_{ext}^2 \epsilon_{int}^2 = \epsilon_{ext}^2 D_{int}/R^2$ and recognize that $P = Q^{-1}$ is a measure of the power needed to maintain free running clock oscillations **Cao et al., 2015** - larger cycles cost more energy per cycle to maintain. Thus, in this case, our trade-off should be understood as one at fixed power.

# Appendix 4

DOI: https://doi.org/10.7554/eLife.37624.014

## Supplementary Methods

Modeling external noise (weather fluctuations) We generate a square wave of period 24 hours to model the day-night cycle of light on Earth with the day length of 12 hr. However, such a square wave is modulated by weather fluctuations, for example, periods of reduced intensity due to passing clouds during the daytime. We model such fluctuating intensity as follows. We assume each weather condition lasts a random interval of time drawn from an exponential distribution of mean $2.4$ hrs (1/10 of a day). During a given weather condition, we set the intensity of light to a random value, drawn uniformly from $[1 - \text{noise}_{\text{ext}}, 1]$ where $\text{noise}_{\text{ext}}$ is the strength of the external noise: 0 means no external noise and 1 means full external noise. This random value will range from 0 to 1 where one represents the maximum intensity during the day. (At night, the intensity is held at zero with no fluctuations). In the simulation of our limit cycle model, we set $\text{noise}_{\text{ext}}$ to 1. However, in our simulations for eight different models of biological clocks, $\text{noise}_{\text{ext}}$ ranges from 0.5 to 1 because when $\text{noise}_{\text{ext}}$ is too high, the system may not get entrained due to the difference in the natural and driving frequencies.

This noisy external signal is coupled to the diverse range of systems studied here in different ways as described in the respective sections. For each system, we simulate a population of organisms where each individual is subject to a different realization of the weather conditions described above. (a) In the Kai clock, the light signal is taken to affect the cellular ATP levels. (b) In the other eight diverse oscillators in the main paper, we coupled the light signal to the parameter specified as coupled to external signals in the original publications. (c) For the dynamical systems model, we assume that the position of the limit cycle is moved by the light signal. When the light intensity is reduced during the day to a value $\rho \in [0, 1]$, we switch the dynamics to an alternative limit cycle (or point attractor) at a fractional distance $\rho$ between the ideal day and night cycles. For example, assume the night cycle is centered at $(0, 0)$ and the day cycle is centered at $(-L, 0)$. During a weather condition with intensity $\rho \in [0, 1]$, we follow dynamics due to a limit cycle located at $(-\rho L, 0)$. We follow the same rules for the point attractor.

Note that we have used a square wave to approximate the natural cycle of light on earth. The square wave also allows for an intuitive derivation of *Equations 1, 2* by dividing up the day-night cycle into four parts: diffusion during the day and during the night, contracting variance during dawn and dusk. For other waveforms, such a clear separation is not possible and all these processes occur concurrently. However, *Equations 1, 2* are expected to still hold up to $O(1)$ prefactors. Numerically, we tested sinusoidal inputs and verified our trade-off relationship.

Modeling internal noise The internal noise represents any source of stochasticity intrinsic to a single cell that would exist even in constant conditions. Such noise could be due to finite copy numbers of molecules, bursty of transcription etc.

We model internal noise in the Kai clock using explicit Gillespie simulations at finite copy number $N$ as described in the section on Kai clocks. For the diverse other biochemical clocks studied here, we add Langevin noise to the dynamical equations, following the prescriptions laid out in the original publications when available. In the dynamical systems models, we model internal noise by adding Langevin noise to the dynamical equations as described in the section on Langevin noise. Each individual particle in our simulation is subject to an independent random realizations of such Langevin noise. In order to ensure an apples-apples comparison between different clocks, we define the strength of the internal noise $\epsilon_{int}$ to be the phase diffusion constant in undriven condition. See Appendix 3 on Dynamical systems simulations for more.

## Measures of clock time-telling quality

We develop and use two distinct measures of performance of noisy clocks driven by noisy inputs.

Mutual information: The performance of the clock is quantified by the mutual information between the clock state $\vec{c}$ and the time $t$,

$$MI(C;T) = \sum_{\vec{c} \in C, t \in T} p(\vec{c}, t) \log_2 \left( \frac{p(\vec{c}, t)}{p(\vec{c}) p(t)} \right) \tag{22}$$

for all $\vec{c}$ in the set of available positions $C$ and all $t$ in the available time bins $T$. (In the dynamical systems model, $\vec{c}$ represents the position in the 2d $r, t$ plane. For the explicit KaiABC biomolecular model, $\vec{c}$ represents the phosphorylation state of KaiC.) We simulate a population of clocks, where each clock is subject to a different realization of input signals, representing different weather conditions and also subject to different realizations of internal Langevin noise (or Gillespie fluctuations). We then collect the trajectories of each clock on the last day of the simulations and calculate the probability distribution $p(\vec{c}|t)$ of clock states at a given (objective) time $t \in [0, 24]$ hrs of the last day in the simulation. The probability function $p(\vec{c})$ is calculated by accumulating the distribution of $p(\vec{c}|t)$ over time $t \in [0, 24]$ hrs of the last day. The position $\vec{c}$ and time $t$ are binned into different bins depending on their values. We start the minimum and maximum values of the bins to the minimum and maximum values of the variables. The bin size in the time dimension is 0.48 hr or 28.8 min, while The bin size in the x and y dimensions are both 0.01.

We refer to this mutual information measure as 'Precision' in *Appendix 1—figure 1d*, *2* and *6a* from the main text.

Population variance along direction of motion: Mutual information is a good indicator of how well the clock encodes information about time. However, it is calculated for the entire day. Often, we want to see how the time-telling ability of a clock changes during the day (e.g., day vs night or before and after dusk). Hence we develop a new measure, closely related to mutual information, but can be computed at specific times of day.

Intuitively, the mutual information quantifies how much the population distributions of clock states at different times $t$ overlap. If these distributions are not overlapping, the clock state is a good readout of the time $t$. Such distributions are shown in *Figures 4b* and *5b* (purple) in the main text.

We argue that only the spread of the clock distribution along the direction of motion of the clock in state space affects mutual information. The spread of the distribution in orthogonal directions does not affect mutual information as much.

To see this, we write mutual information between clock state $\vec{c}$ and time $t$ as,

$$MI(C;T) = H(T) - H(T|C). \tag{23}$$

Here $H(T)$ is a constant, independent of the clock mechanism. Thus, *MI* depends entirely on the entropy of the distribution $p(t|c)$ of real times given clock state $c$, averaged over different clock states,

$$H(T|C) = \int p(c) dc\, H(T|c) \tag{24}$$

$$= -\int p(c) dc \left[ \int dt\, p(t|c) \log p(t|c) \right] \tag{25}$$

Consider a clock whose state-space is two dimensional with a periodic x-axis as shown in *Appendix 4—figure 1*. Further, assume that the distribution $p(\vec{c}|t)$ of clock states at a given time is supported on a rectangle of size $a_x \times a_y$ as shown in *Appendix 4—figure 1* and that the clock states move along the x-axis at a uniform velocity $u$. This situation implies that

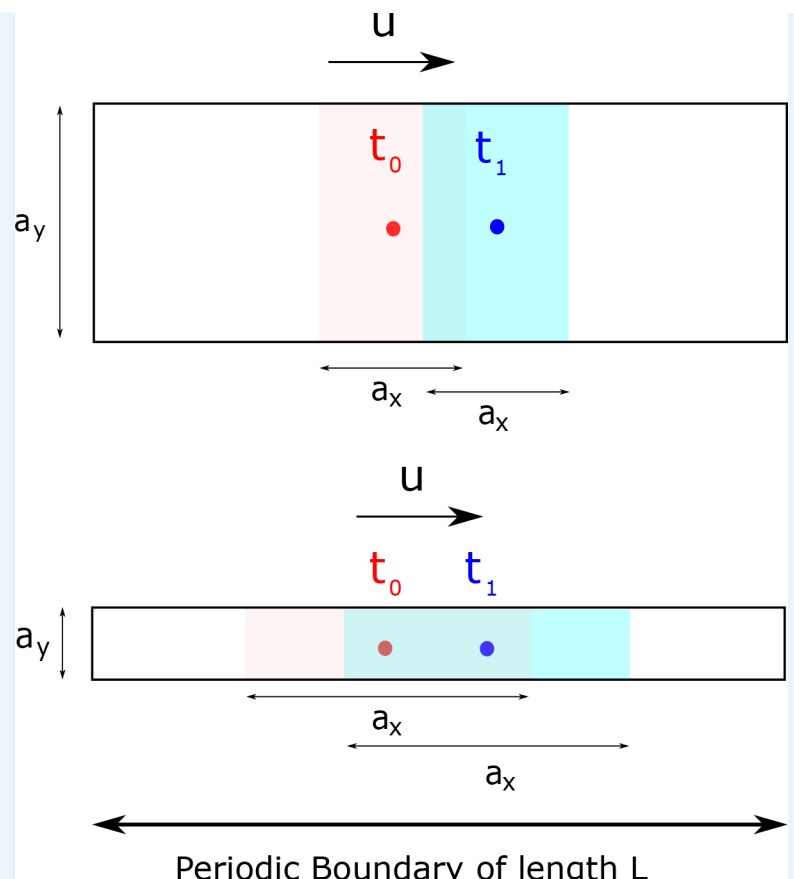

**Appendix 4—figure 1.** Mutual information $MI(\vec{c}, t)$ between clock state $\vec{c}$ and time $t$ is only affected by the variance of the clock state distribution $p(\vec{c}|t)$ at a given time $t$ along the direction of motion and not orthogonal to it. In this toy example, we assume the distribution $p(\vec{c}|t)$ to be supported on a rectangle of size $a_x$ and $a_y$ in a 2d clock state space. The clock state moves at a speed $u$ in the x-direction. Time telling quality is affected by how much the population at different times overlap with each other. Consequently, clocks with large $a_x$ and small $a_y$ (*bottom*) have lower mutual information $MI(\vec{c}, t)$ relative to clocks with small $a_x$ and large $a_y$ (*top*). Consequently, we use the population variance along the direction of motion as an instantaneous measure of time-telling ability in the paper.

DOI: https://doi.org/10.7554/eLife.37624.015

$$p(t|c) = \begin{cases} 0 & \text{for } |c_x - ut| > a_x \\ \frac{u}{2a_x} & \text{for } |c_x - ut| \leq a_x \end{cases}$$

So,

$$\begin{aligned} H(T|C) &= -\int p(c)dc \int_{t=(c_x-a_x)/u}^{(a_x+c_x)/u} dt \frac{u}{2a_x} \log\left(\frac{u}{2a_x}\right) \\ &= \log\left(\frac{2a_x}{u}\right) \end{aligned}$$

Since $MI(C;T) = H(T) - H(T|C)$, *MI* depends on $-\log a_x$ and is independent of $a_y$, meaning that only the spread in the direction of motion $a_x$ affects the mutual information. Consequently, to understand the quality of time-telling at different times of the day, we project the population variance of $p(\vec{c}|t)$ to the direction of the instantaneous velocity of the center of mass of $p(\vec{c}|t)$.

## Cramer-Rao bounds

Cramer-Rao (CR) bounds quantify the total available information about phase in a given length of history of the signal. Any estimator working with that length of history must necessarily have higher variance (i.e., higher error) than the Cramer- lower bound corresponding to that length of history. In the limit of infinitely long histories, the CR bound in this context corresponds to zero error; with any finite binning in time, the upper bound on MI is simply set by the number of bins in time. In our case, this bound is given by $log_2 50 = 5.64$ bits. As shown in the main paper, as $L/R \rightarrow 0$, limit cycles process longer and longer histories of the external signal. Consequently, the mutual information for such cycles approaches the upper bound in the limit $L/R \rightarrow 0$ (assuming no internal noise) when computed with the same number of temporal bins (50 in this case).

# Appendix 5

DOI: https://doi.org/10.7554/eLife.37624.016

## Circle Map - Dark pulse phase shift

During the daytime, sunlight intensity fluctuates because of cloud cover and we have referred to these fluctuations as external noise. In our simulations, we subject each individual in a population to a different realization of these weather conditions and compute the resulting population variation of clock state. Such variation limits the ability of the cell to read out the objective time from the clock state.

Here, we relate the population phase variance caused by random cloud cover in our dynamical systems model to the geometrically computed Phase Response Curve (PRC) due to a single dark pulse administered during the day. Using this geometric method, we will find that the ability of limit cycle to withstand external intensity fluctuations increases with $R/L$, the size $R$ of limit cycles relative to their separation $L$. In particular, we will show geometrically that the gain in phase variance during the day $\sigma^2 \Rightarrow \sigma^2 + \sigma^2_{clouds}$ scales as $(L/R)^2$, in perfect agree with stochastic weather simulations.

To compute the scaling relationship of $\sigma^2_{clouds}$, we compute the phase shift $\Delta\Phi$ caused by a single dark pulse with width $\tau$ on the limit cycles with angular speed $\omega$ (i.e., the Phase Response Curve (PRC) corresponding to such a dark pulse). **Appendix 5— figure 2a** shows an example of a dark pulse in the signal and how it affects the trajectory. Consider a clock at state $\theta$ on the day cycle. A dark pulse of length $\tau$ administered just then will change the dynamics to that of the night cycle. This clock has state $\phi = P(\theta)$ with respect to the night cycle and will evolve for a time $\tau$ according to the night cycle dynamics, reaching a new state $\phi + \omega\tau$, at a radial position determined by $R, L$. At the end of the dark pulse, we use the night-day circle map, $\theta = Q(\phi)$, to find the clock state back on the day cycle. Note that all these shifts depend on the limit cycle geometry, that is, on $R$ and $L$, as shown in **Appendix 5— figure 2**. We can write each mapping using simple trigonometry:

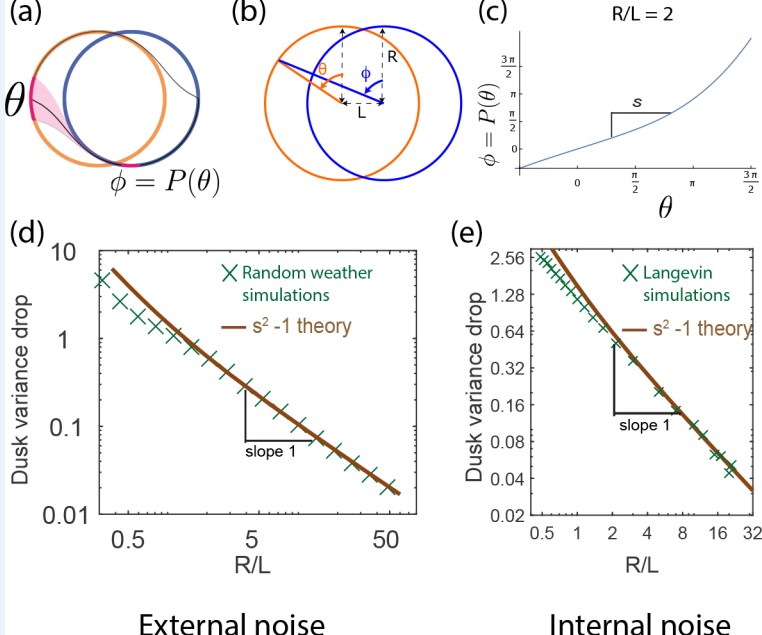

**Appendix 5—figure 1.** The population variance of clock states is reduced by dusk and can be computed geometrically. (**a**) A population of clocks near state $\theta$ on the day cycle is mapped to the neighborhood of state $\phi$ on the night cycle by the dusk transition. We define $\phi = P(\theta)$ to be the map relating the clock state $\theta$ on the day cycle just before dusk to its eventual position $\phi$ on the night cycle after dusk (assumed greater than the relaxation time). (**b**) This map can be

analytically computed for circles of size R with centers separated by length L. (**c**) For a given R/L = 2 , we obtain $P(\theta)$ shown here. Since $\theta = \pi/2$ corresponds to the dusk time of the entrained trajectory, the slope $s^{-1} = dP/d\theta$ at $\theta = \pi/2$ determines the change in population variance of clock states at dusk. (**d,e**) The variance drop $s^2$ at dusk, defined as $\sigma^2 \to \sigma^2/s^2$ at dusk, seen in both the external (averaging over weather) and internal noise (averaging over Langevin noise) simulations agree well with the geometrically computed $s(R/L)$, especially at large $R/L$. We find that $s^2 - 1 \sim L/R$ for large-$R/L$ limit cycles.

DOI: https://doi.org/10.7554/eLife.37624.017

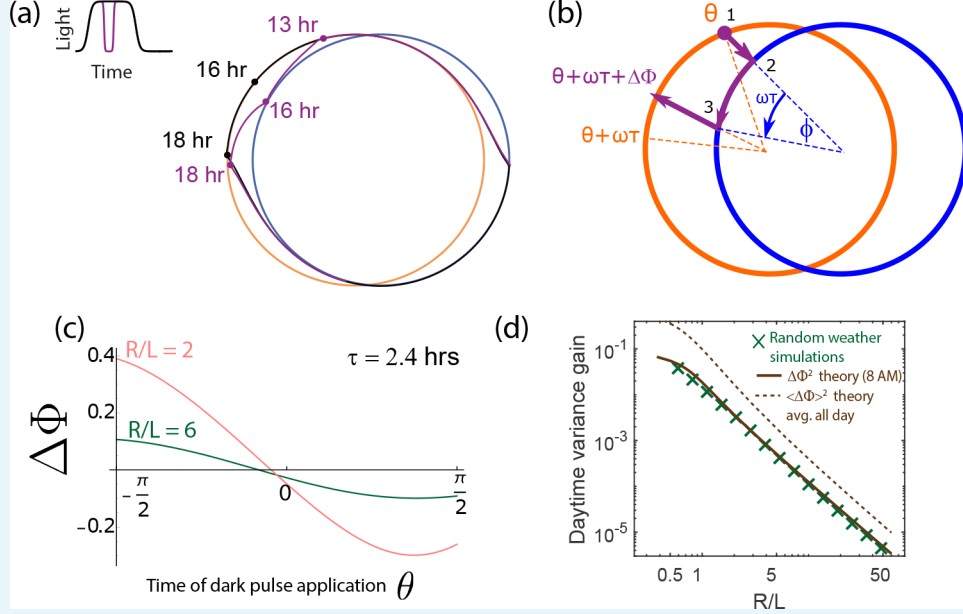

**Appendix 5—figure 2.** Increase in population variance due to random weather conditions can be estimated from the phase shifts $\Delta\Phi$ due to dark pulses (i.e., the Phase Response Curve). (**a**) A single dark pulse administered during the day shifts the phase of a clock (purple) relative to a clock that experiences no such dark pulse (black). (**b**) We can compute the phase shift $\Delta\Phi$ due to such a dark pulse geometrically by computing the deviation in trajectory. Assuming a dark pulse of length $\tau$, the clock evolves for a time $\tau$ according to the night cycle dynamics. At the end of such a pulse, we switch back to the day limit cycle and compute the resulting phase shift $\Delta\Phi$. (**c**) The resulting phase shift $\Delta\Phi$ due to a pulse of length $\tau = 2.4$ hrs, depends on the time $\theta$ when it is administered but is generally smaller for larger $R/L$. (**d**) We find that $\Delta\Phi^2$ for a specific $\tau = 2.4$ hrs dark pulse administered at the same time (8 AM) falls as $(L/R)^2$ for large-$R/L$ limit cycles. This trend matches the variance gain $\sigma^2_{clouds}$ seen in stochastic simulations that average over random weather conditions (pulses of different length, intensity and time of application). The broken brown curve shows a theoretical prediction for such an average $\langle\Delta\Phi^2\rangle$, obtained by sampling the curve shown in (**c**) at different points of application and differing intensity. Despite the presence of a variance-reducing zero around mid-day in (**c**), $\sigma^2_{clouds}$ drops as $(L/R)^2$, much as $\Delta\Phi^2$ for any particular pulse. (Brown theory curves translated together using one fitting parameter).

DOI: https://doi.org/10.7554/eLife.37624.018

$$\phi = P(\theta) = \arctan\left(\frac{L + R\sin\theta}{R\cos\theta}\right) \tag{26}$$

and

$$\theta^* = Q(\phi) = \arctan\left(\frac{-L + R\sin(\phi + \omega\tau)}{R\cos(\phi + \omega\tau)}\right). \tag{27}$$

Notice the mapping $Q$ only differs from $P$ by changing $L$ to $-L$. We also include the diagram showing the transition due to dark pulse in **Appendix 5— figure 2**. The process '1' corresponds to $\phi = P(\theta)$, '2' corresponds to the rotation on the night cycle $\phi \to \phi + \omega\tau$, and '3' corresponds to the transition back to the day cycle $\theta^* = Q(\phi + \omega\tau)$. Combining this three processes, we write $\theta^*$ as $\theta^*(\theta, \tau, L/R)$ and expand it in the limit that $L/R \Rightarrow 0$ to obtain that

$$\Delta\Phi = -\frac{L}{R}(\cos(\theta + \omega\tau) - \cos(\theta)) + \mathcal{O}\left(\frac{L}{R}\right)^2 \tag{28}$$

where $\Delta\Phi = \theta^* - (\theta + \omega\tau)$ because $\theta + \omega\tau$ is the phase of the clock if it did not experience the dark pulse.

This expression $\Delta\Phi$ indicates the amount of phase shifted that the cloud causes. With different clocks experiencing different weather conditions, the variance gained among the population due to the fluctuation of sunlight grows like $|\Delta\Phi|^2 \sim (L/R)^2$. We see good agreement between stochastic weather simulations and this geometric computation as shown in **Appendix 5— figure 2d**.

In this calculation, we focused on dark pulses administered at a fixed generic time (8 AM in **Appendix 5— figure 2d**). However, the PRC $\Delta\Phi(\theta)$ for dark pulses has a zero at a specific time of the day (see **Appendix 5— figure 2c**). That is, for each dark pulse of width $\tau$, there exists a time of administration such that $\Delta\Phi = 0$! In fact, such a dark pulse has an entraining effect, reducing the population variance. We leave experimental and theoretical investigation of the counter-intuitive effects of such specially timed dark pulses to future work.

Here, we show that even if we include such dark pulses with an entraining effect, the variance gained at the end of the day is still proportional to $(L/R)^2$ in the limit that $L/R$ goes to zero. To simplify our derivation but retain the essence of what dark pulses do during the daytime, let's us consider dark pulses coming at three times: in the morning ($\theta = -\pi/2$), around noon ($\theta = -\omega\tau/2$ with small $\omega\tau$), and in the evening ($\theta = \pi/2$). Starting the day with variance $\sigma_0^2$, by the end of the day the variance becomes

$$\sigma^2 = \frac{\sigma_0^2 + (\Delta\Phi)_{\theta=-\pi/2}^2}{\left(1 + \left(\frac{d\Delta\Phi}{d\theta}\right)_{\theta=-\omega\tau/2}\right)^2} + (\Delta\Phi)_{\theta=\pi/2}^2 \tag{29}$$

$$\approx \frac{\sigma_0^2 + \left(\frac{L}{R}\sin\omega\tau\right)^2}{\left(1 + \frac{2L}{R}\sin\left(\frac{\omega\tau}{2}\right)\right)^2} + \left(\frac{L}{R}\sin\omega\tau\right)^2 \tag{30}$$

$$\sigma^2 \approx \sigma_0^2 + 2\left(\frac{L}{R}\sin\omega\tau\right)^2 + \mathcal{O}\left(\frac{L}{R}\right)^2. \tag{31}$$

Thus, the variance gained due to fluctuation, $\sigma^2 - \sigma_0^2 = \sigma_{clouds}^2$, is proportional to $(L/R)^2$. This simple derivation may not rigorously reflect the correct constant in front of $(L/R)^2$ term, but the full rigorous derivation, concerning the dark pulses coming randomly at random time during the day, should yield the same power law dependent on $L/R$. **Appendix 5— figure 2d** shows that averaging $\Delta\Phi^2$ over pulses administered at different times numerically (dashed line) results in the same power law as for single pulses and as seen in stochastic weather simulations.

## Circle Map - Step Response Curve

In our main paper, we claim that the variance of the clock state across a population drops $\sigma^2 \Rightarrow \sigma^2/s^2$ at dusk where $s^2 - 1 \sim L/R$ as $L/R \Rightarrow 0$. Data from Langevin simulations was presented. Here we will derive this result using a simple geometric argument about circle maps.

We define $\phi = P_T(\theta)$ to be the phase on the night cycle that a clock evolves to, after time a time $T$, if the lights were suddenly turned off when the clock is at state $\theta$ on the day cycle. See **Appendix 5—figure 1a,b**. In principle, with complex relaxation dynamics between the limit cycles, $P_T(\theta)$ could show complex dependence on $T$. However, we work in a simplified model where the angular frequency of the clock is independent of the amplitude of oscillations. In this limit, $T$ only causes an overall shift in $\phi = P_T(\theta)$; that is, we can write $P_T(\theta) = P(\theta) + \omega T$ where $\omega$ is the angular frequency of the clock. In what follows, we will be interested in the derivative of $\partial_\theta P_T(\theta)$; hence we will work with $P(\theta)$ instead of $P_T(\theta)$.

This circle map, $\phi = P(\theta)$, is important since it determines whether two differing day-time clock states are brought closer or taken further at dusk and thus determines the rate of entrainment of a population to the external signal. Consider two organisms that have nearby but distinct clock states $\theta_0$, $\theta_0 + \Delta\theta$ at dusk. After dusk, these two clocks will be mapped to $P(\theta_0)$ and $P(\theta_0 + \Delta\theta) \approx P(\theta_0) + \Delta\theta\, dP(\theta) d\theta|_{\theta=\theta_0}$ respectively. Thus, dusk changes the difference between the clock states from $\Delta\theta$ to $\Delta\phi$ where,

$$\Delta\phi \approx \Delta\theta \frac{dP(\theta)}{d\theta}\bigg|_{\theta=\theta_0} \tag{32}$$

By a similar argument, if the phase variance of clock states across a population is $\sigma^2$ before dusk, it will be reduced by,

$$\sigma^2 \xrightarrow{dusk} \sigma^2 \left( \frac{dP(\theta)}{d\theta}\bigg|_{\theta=\theta_0} \right)^2 \tag{33}$$

This expression is valid in the regime where the population variance $\sigma^2$ is small enough to linearize the circle map $P(\theta)$. Similar considerations apply to the dawn transition between the night and day cycle as well. Both circle maps were recently experimentally characterized for *S. elongatus* in **Leypunskiy et al. (2017)**.

In our simple theoretical model where clock frequency does not change with amplitude (i.e. the radial coordinate), we can easily compute $P(\theta)$ from geometry. In **Appendix 5—figure 1**, we draw a diagram of the transition from a particle on the day cycle at the phase $\theta$ to the night cycle at the phase $\phi$. By trigonometry, we write

$$\phi = P(\theta) = \arctan\left( \frac{L + R\sin\theta}{R\cos\theta} \right), \tag{34}$$

and derive

$$s^2 - 1 = \left( \frac{dP(\theta)}{d\theta} \right)^{-2} - 1 \tag{35}$$

$$= \frac{L(2L^3 + 7LR^2 - 3LR^2\cos(2\theta) + 4R(2L^2 + R^2)\sin\theta)}{2R^2(R + L\sin\theta)^2} \tag{36}$$

$$= 2\sin(\theta)\frac{L}{R} + \mathcal{O}\left( \frac{L}{R} \right)^2, \tag{37}$$

where $\theta$ corresponds to the angle on the day cycle at dusk, which is at $\pi/2$ in **Appendix 5—figure 1a**. This equation implies that as the day and night limit cycle gets closer, the geometric focusing effect $s$ converges to one. This asymptotic behavior is intuitive because if $L = 0$, meaning no transition, then the variance should remain the same ($s = 1$, so $\sigma^2 \to \sigma^2/1^2$ at the transition).

Remarkably, our geometric derivation of $s^2 - 1$ matches the variance drop $\sigma^2 \to \sigma^2/s^2$ seen in stochastic simulations of weather conditions; see **Appendix 5—figure 1b**. The variance gain during the day is the result of the fluctuation of sunlight, simulated as random dark pulses of random intervals, amplitude and time of delivery. Such variance is accumulated during the day and the drop over dusk time is measured (green Xs).

**Appendix 5—figure 1e** shows the variance drop seen in simulations with internal noise in Langevin simulations. While the cause of variance increase during the day is different (finite

copy number effects), the variance drop at dusk agrees well with the geometric computation of $s^2$ and thus with the external noise simulations as well. In both cases, the simulations and geometric theory show that $s^2 - 1 \sim L/R$ as $L/R \Rightarrow 0$.

## Appendix 6

DOI: https://doi.org/10.7554/eLife.37624.019

# Langevin model of finite copy number fluctuations

Chemical reactions that occur in the bulk of a homogeneous solution can be described by a set of ordinary differential equations. However, within a single cell the copy number of molecule is limited and thus the reaction carries internal noise from the stochastic fluctuations. Gillespie showed that chemical reactions under finite copy number can be approximated by a Langevin dynamics using the following argument (**Gillespie, 2007**),

Consider an elementary reaction

$$A + B -> C + D \tag{38}$$

with the forward rate constant $k_+$, during each infinitesimal time $\delta t$, the probability of the occurrence of this reaction follows a Poisson distribution whose mean and variance both equal to $R_+ \delta t = k_+ \cdot N_A \cdot N_B \cdot \delta t$. Integration over a larger time step, the Poisson distribution can be approximated into a Gaussian form, resulting in Langevin dynamics,

$$dN_A = -k_+ \cdot N_A \cdot N_B \cdot dt + \sqrt{R_+} dW \tag{39}$$

where $W$ is a standard Wiener process of mean 0 and autocorrelation function $\langle W(t_1)W(t_2)\rangle = \delta(t_1 - t_2)$.

To describe a chemical reaction network, the Langevin equation for each species consists of contributions to the noise from each reaction where the species is involved. Now consider adding another reaction

$$C + D -> A + B \tag{40}$$

with the rate constant $k_-$, then the Langevin equation for species A becomes,

$$dN_A = -k_+ \cdot N_A \cdot N_B \cdot dt + k_- \cdot N_C \cdot N_D \cdot dt + \sqrt{R_+} dW_1 + \sqrt{R_-} dW_2 \tag{41}$$

where $R_+ = k_+ \cdot N_A \cdot N_B$ and $R_- = k_- \cdot N_C \cdot N_D$ respectively denote the number rates of the forward and the backward reaction; $dW_1$ and $dW_2$ are identical independent standard Wiener processes.

To fully determine the effect of the noise using the Langevin dynamics for a chemical reaction network, one needs to consider all of the reactions corresponding to the species of interest; the noise term usually becomes time-dependent and multiplicative. To simplify the description of internal noise in our phenomenological model of limit cycle/point attractor, we take a first order approximation that the diffusion coefficient in the reaction coordinate space is homogeneous in both space and time. (See similar treatments of another biological system in **Potoyan and Wolynes (2014)**. In contrast, our explicit KaiABC simulations, as well as numerical simulations on the other types of bio-oscillators, presented later, do not make this simplifying assumption of homogeneous diffusion.) This allows us to write a 2-dimension phenomenological stochastic differential equation

$$d\vec{z} = f(\vec{z}, t) \cdot dt + \sqrt{2D} \cdot d\vec{W} \tag{42}$$

where the $f(\vec{z}, t)$ denotes the deterministic dynamics driven by day-night cycles and the diffusion constant $D$ is assumed to be inversely proportional to the total number of Kai-C molecules within the cell. For limit cycles of radius $R$, we set $D \sim R^2 \epsilon_{int}^2$. Then, $\epsilon_{int}^2$ is the diffusion constant for the *phase* of the oscillator. We hold $\epsilon_{int}^2$ fixed while changing $R$ to make a fair comparison across systems of different size.

## Population variance

For the cell to carry out a reliable computation, the population variance from the internal noise needs to be reduced. Such noise reduction comes from the dynamics of the attractor. In the limit cycle attractor mechanism, the internal noise reduction is performed only along the radial axis but not along the flat attractor direction.

In contrast, the point attractor mechanism is able to limit population variance due to internal noise in all directions due to the effective 'curvature' of the dynamics. Here we analytically estimate the steady-state population variance for a point attractor mechanism. The population variance is together determined by the diffusive term $\sqrt{2D} \cdot d\vec{W}$, and the noise reduction effect from the restoring force of the point attractor's harmonic well. During each infinitesimal time $\delta t$, the internal noise increase the variance by

$$\sigma^2(t+\delta t) = \sigma^2(t) + 2D\delta t. \tag{43}$$

In contrast, the overdamped deterministic motion within a harmonic well provides a focusing effect that reduces the variance exponentially with time. To quantify this focusing effect, consider a 1-d overdamped dynamics of a particle within a harmonic energy well of $V(r) = k \cdot r^2$. The solution to the equation of motion is $r(t) = r_0 \cdot e^{-2kt}$, with initial position $r(0) = r_0$. Consider an ensemble of points with a mean initial position $\mu_0$ and a initial variance of $\sigma_0^2$, one can solve the dynamics of the mean as

$$\mu(t) = \mu_0 \cdot e^{-2kt} \tag{44}$$

and the dynamics of the variance as

$$\sigma^2(t) = \sigma_0^2 \cdot e^{-4kt} \tag{45}$$

Thus, per infinitesimal time $\delta t$, the geometric focusing effect of the energy well of the point attractor reduces the population variance by

$$\sigma^2(t+\delta t) = \sigma^2(t)/g \tag{46}$$

where $g = e^{4k\delta t}$.

Under the competition between the spreading effect from the internal noise and the geometrical focusing effect from the deterministic dynamics, the population variance reaches a steady value solved by

$$\sigma_{st}^2 = \frac{\sigma_{st}^2 + 2D\delta t}{g} = \frac{\sigma_{st}^2 + 2D\delta t}{e^{4k\delta t}} \tag{47}$$

and by taking the limit of $\delta t$ goes to 0, we have $\sigma_{st}^2 = D/2k$.

