## [Decision Letter]

Thank you for submitting your article "Biophysical clocks face a trade-off between internal and external noise resistance" for consideration by *eLife*. Your article has been reviewed by three peer reviewers, and the evaluation has been overseen by a Reviewing Editor and Naama Barkai as the Senior Editor. The following individual involved in review of your submission has agreed to reveal his identity: Sandeep Krishna (Reviewer #2).

The reviewers have discussed the reviews with one another and the Reviewing Editor has drafted this decision to help you prepare a revised submission.

This paper addresses the question of noise in circadian clocks from a theoretical point of view, contrasting two main strategies for telling time: a free-running entrained clock (two limit cycles), and a set of two attractors, one for the day and one for the night (two attractors). The main insight is that these two strategies differ in how they can absorb external vs. internal noise. The limit cycle strategy is robust to external noise, but not to internal one. The opposite is true for the attractor strategy. This finding is consistent with experimental observations in bacteria.

All of the reviewers and the editor agreed that the paper develops an elegant and interesting theory, which is critically tested. All the reviewers praised the manuscript. The reviewers raised a number of comments (both regarding the work and the presentation) that should be addressed before publication.

*Reviewer 1:*

Evolutionarily speaking, I would expect the hourglass model to be the ancestral state, so it is not clear that, say, in *P. marinus*, it is really adaptive not to have a (more evolved) clock. A convincing argument could be that in some contexts a clock "de-evolved" towards an hourglass model. It would be good if the authors could comment on this.

*Reviewer 2:*

1) The approximation the authors make in writing Langevin equations needs to be explained better, it's very unclear right now. To be specific, in the one case where they show a Langevin equation for a circadian oscillator (Appendix 2 Equation 5), I'm fine with Appendix 2 Equations 6 and 7. I assume the simulations use this matrix and their simulations for the other oscillators also use a similar matrix (if so, the authors should say this explicitly, and if not, then the following worry holds for these simulations too). Similarly, Appendix 6 Equation 41 (Gillespie's previous work) is fine too. But I don't understand exactly how the authors get Appendix 6 Equation 42 because for f(z→,t) they say they use the original nonlinear form from the purely deterministic equations. One way to get Appendix 6 Equation 42 from Appendix 6 Equation 41 (or whichever other deterministic equation set one starts with) would be to linearize Appendix 6 Equation 41 around some (fixed) point N*_A, N*_B – in the linear regime I can see how the noise terms R+ and R- would reduce to the noise term in Appendix 6 Equation 42 but then f(z→,t) would also be linear in (N_A-N*_A), (N_B-N*_B) etc. I don't see any rigorous way to keepf(z→,t) as the same nonlinear function of N_A, N_B, etc., without also keeping R+ and R- to be the nonlinear functions of N_A and N_B that they are in Appendix 6 Equation 41. Ideally, I would like to see a rigorous derivation, say where Appendix 6 Equation 42 arises from Appendix 6 Equation 41 as some small parameter is sent to zero. If that's not possible then perhaps the authors can at least compare this approximation with the full Gillespie simulation for the *S. elongatus* and *P. marinus* oscillators they show in Figure 1 of the main text. Since they have the full Gillespie simulation there, it should be easy to write the Master equation, derive the nonlinear Langevin equation (akin to Appendix 6 Equation 41) and then make exactly the same approximation. I would like to see in such a case where (i.e., for what parameter values in the model) the approximation gives results that match the Gillespie simulation and where it doesn't (the latter is important – it's very useful for a reader to know where the approximation fails; e.g., does it fail say for very spiky oscillations where one variable spends a lot of time near zero but periodically rises to a high value?). My expectation is actually that the results will not be too different, so I don't anticipate that this will invalidate any of the trends the authors see, but it's important to check this.

2) The authors have used a particular type of external noise, namely they add noise to the amplitude of a square pulse. As far as I can tell there's no variability in the time at which the square pulse drops to zero or the time when it rises (let me call such variability "noise in the phase of the square wave"). (If I have misunderstood and their external noise does include this then what is written below can be ignored, but the authors could try to be more clear in their description of the external noise.) Their argument in the dynamical systems section makes me think that such noise in the phase of the square wave would cause the limit cycle oscillators to diffuse in the direction tangent to the trajectory, just as they do with internal noise. But I would still call this external noise because it doesn't have a 1/sqrt(N) dependence. So is the distinction really between external and internal noise (which I interpret as noise that is independent of N vs. 1/sqrt(N)), or is it between noise that affects the phase and noise that doesn't, irrespective of whether it has a 1/sqrt(N) dependence? In the circadian case, maybe there is an argument to say that this is just semantics, because external noise due to weather fluctuations won't change the phase of the day-night cycle. I agree with this, but I'm looking at their results beyond circadian clocks alone, because the audience for their paper will include biologists interested in other oscillators too. For instance, NF-κB oscillations may well be driven by periodic cytokine secretions and "external noise" might well perturb the phase as well as the amplitude of the oscillating cytokine. In such a case, if the real difference in behaviour between limit cycles and hourglass clocks is actually in their response to noise that affects the phase of the driving signal vs noise that does not, then the nomenclature external vs internal noise will inadvertently mislead people. I think simply looking at the behaviour of their phenomenological oscillators with noise in the phase of the square wave will clear this up.

3) The authors have come up with some neat measures to compare the oscillators they look at. In particular, I think the precision measure (first shown in Figure 1D) based on mutual information will prove to be very useful for other researchers in a variety of contexts. Having a measure like this is critical for this study because otherwise how do you compare different models with different parameter values, different variables, and perhaps even different dimensions? So I actually think the authors undersell themselves here – the measure and its purpose deserves more than one throwaway line in the main text (I doubt anyone will get anything out of "variance along the most informative directions"). If the authors don't want to bring the mathematics into the main text, at least they should say more about comparing models when they talk about Figure 1D (incidentally, although it's clear where they talk about Figure 1D in the main text, an explicit reference to "Figure 1D" is missing and may be useful for a reader who has looked at that figure first and is trying later to find where it is discussed in the text.)

4) In their dynamical systems models, the limit cycle has a constant amplitude. I'm curious: how much would their argument be affected if the radius R in fact changed a lot over the course of the trajectory? (again I'm thinking of very spiky oscillations, or more complex oscillations with non-planar phase plots, which may not be relevant for circadian systems but may be relevant for other biological oscillations.) Wouldn't a varying R cause a non-zero curvature for the limit cycle? If so, then the timing of the day-night transition and whether it happens where curvature is zero or not may become important. How would a large variation of internal noise along the trajectory, due say to R becoming close to zero, affect their results? I'll leave it to the authors to decide whether any of this is in the scope of the current paper or not, but perhaps a quick discussion of the approximation involved in keeping R fixed on the trajectory is warranted.

5) Whenever the authors talk of the hourglass clocks they present them as having damped oscillations. From their arguments, surely this is not necessary and the system would work as well or as badly if it was not spiraling into the fixed point (eigenvalues having an imaginary component), but rather had an "overdamped" approach (real, negative eigenvalues)? As far as I can tell, as long as the relaxation time is of the same order as the day-night cycle the behaviour should be the same.

6) I found parts of the manuscript hard to follow. I needed to flip back and forth between the main text and supplement a fair bit, which to me indicates that some of the supplement needs to go into the main text, or parts of the main text need more amplification. I already mentioned the too brief discussion of the precision measure. In the main text, it also took me a while to figure out what s was (just before Equation 1) – it would help to define this more clearly. And it would help to call out specific sections of the supplement instead of using the generic "see SI". Within the supplement, why not just write out the Langevin equations soon after Appendix 3 Equation 14 rather than asking the reader to jump to find it? And I also think it would be good to include the diffusion matrix the authors have used for each of the "Other oscillators", so that it is clear that they have not used the approximation in Appendix 6 Equation 42 for these simulations (or have they?).

7) Just out of curiosity – a typical argument made for explaining some aspects of the architecture of circadian oscillators lies in other features such as temperature compensation. Have the authors thought about whether limit cycles or hourglasses do better at temperature compensation?

8) There are a few typos:

Subsection “Limit cycle clocks and point attractor clocks”, second paragraph: diverge should be diverse.

Subsection “Simulations”, fourth paragraph: Langeving should be Langevin

Appendix 6 Equation 39 and Appendix 6 Equation 41: d*N_a_* should be d*NA*.

*Reviewer 3:*

My only concern is with regards to very dense, technical and somewhat cumbersome presentation of the Results. This could be an example where the principle "less is more" could be applicable. The main and most important idea of the paper is clear from Figure 1 and the universal character of the discovered trend shown in Figure 2 is impressive. However, the subsequent analysis in the main text it is very hard to follow and requires full immersion to an appendix to understand. This could be more appropriate for a more specialized journal. Results about trade-off in Figure 6 seems very interesting but presented in a brief passing here.

---

## [Author Response]

Reviewer 1:Evolutionary speaking, I would expect the hourglass model to be the ancestral state, so it is not clear that, say, in P. marinus, it is really adaptive not to have a (more evolved) clock. A convincing argument could be that in some contexts a clock "de-evolved" towards an hourglass model. It would be good if the authors could comment on this.

Phylogenetic analysis by Dvornyk indicates that the last common ancestor of *Prochlorococcus* had an intact KaiABC gene cluster, but that the ancient form of the Kai system likely consisted of KaiB and KaiC before the KaiA gene appeared. This does indeed suggest that the hourglass-like dynamics of *Prochlorococcus* represent a regression to an ancestral state. Though the evolutionary trajectory that gave rise to the first circadian rhythms is not known, it is a fascinating question. The notion that the ancestors of clocks were driven systems that did not free-run is consistent with our theoretical analysis: the performance of non-free running systems are less sensitive to internal noise and to a mismatch between kinetic parameters in the system and the period of the diel cycle in the environment, features that were likely present in the earliest Kai systems.

Our analysis shows a trade-off between the strength of random fluctuations in the environment and internal noise in favoring free-running oscillations. Both of these factors may have contributed to push *Prochlorococcus* towards giving up a free-running clock. *Prochlorococcus* is typically found in the open ocean near the equator, where the external environment may be very regular. Further, there is an overall trend in *Prochlorococcus* towards reduced genome size and cell size which likely imply increased internal noise.

As we now say in the Discussion:

“At the low protein copy numbers such as those found in *P. marinus*, damped clocks keep time more reliably than free running clocks. […] In addition to the noisy internal environment of *P. marinus*, the external environment might also play a role in selecting a damped clock; *P. marinus* is typically found in the open ocean, where the external environment may be more regular than the fresh water habitat of *S. elongatus.*”

Reviewer 2:1) The approximation the authors make in writing Langevin equations needs to be explained better, it's very unclear right now. To be specific, in the one case where they show a Langevin equation for a circadian oscillator (Appendix 2 Equation 5), I'm fine with Appendix 2 Equations 6 and 7. I assume the simulations use this matrix and their simulations for the other oscillators also use a similar matrix (if so, the authors should say this explicitly, and if not, then the following worry holds for these simulations too).

We appreciate the reviewer’s comment and we have revised the appendix of Langevin equations for better clarity.

Indeed, Appendix 2 Equations 6 and 7 were from an early paper of Goldbeter which carefully derived the form of this ‘diffusion matrix’ – i.e., precise form of the Langevin equation – starting from deterministic equations. The derivation carefully accounts for the physical dimensions of various equations and thus derives the correct of the Langevin noise term.

All other models being simulated using Langevin equations derived following Goldbeter’s work that led to Appendix 2 Equations 6 and 7. Our revised appendix makes this explicit.

*Similarly, Appendix 6 Equation 41 (Gillespie's previous work) is fine too. But I don't understand exactly how the authors get Appendix 6 Equation 42 because for* f(z→,t) *they say they use the original nonlinear form from the purely deterministic equations. One way to get Appendix 6 Equation 42 from Appendix 6 Equation 41 (or whichever other deterministic equation set one starts with) would be to linearize Appendix 6 Equation 41 around some (fixed) point N*_A, N*_B – in the linear regime I can see how the noise terms R+ and R- would reduce to the noise term in Appendix 6 Equation 42 but then* f(z→,t) *would also be linear in (N_A-N*_A), (N_B-N*_B) etc. I don't see any rigorous way to keep*f(z→,t) *as the same nonlinear function of N_A, N_B, etc., without also keeping R+ and R- to be the nonlinear functions of N_A and N_B that they are in Appendix 6 Equation 41. Ideally, I would like to see a rigorous derivation, say where Appendix 6 Equation 42 arises from Appendix 6 Equation 41 as some small parameter is sent to zero.*

We have cleared up the cause for this misunderstanding; Appendix 6 Equation 42 was meant to describe only for the simplified 2-dimensional dynamical systems theory of limit cycles and *not* for the eight explicit biochemical models. For the latter, we did indeed use the correct form of the equations (as derived e.g., by Goldbeter) discussed earlier.

We have revised the text in that Appendix to read,

“…our explicit KaiABC simulations, as well as numerical simulations on the other types of bio-oscillators, presented later, do not make this simplifying assumption of homogeneous diffusion.”

If that's not possible then perhaps the authors can at least compare this approximation with the full Gillespie simulation for the S. elongatus and P. marinus oscillators they show in Figure 1 of the main text. Since they have the full Gillespie simulation there, it should be easy to write the Master equation, derive the nonlinear Langevin equation (akin to Appendix 6 Equation 41) and then make exactly the same approximation. I would like to see in such a case where (i.e., for what parameter values in the model) the approximation gives results that match the Gillespie simulation and where it doesn't (the latter is important – it's very useful for a reader to know where the approximation fails; e.g., does it fail say for very spiky oscillations where one variable spends a lot of time near zero but periodically rises to a high value?). My expectation is actually that the results will not be too different, so I don't anticipate that this will invalidate any of the trends the authors see, but it's important to check this.

As mentioned above, our simulation for the KaiABC and other bio-oscillators do *not* involve such an approximation; we have now revised our presentation to make this clear. Also as the reviewer expected, full simulations without these approximations from our manuscript showed similar results with the simplified 2-dimensional models.

We thank the reviewer for helping clear up our presentation of approximations and assumptions in the simulations.

2) The authors have used a particular type of external noise, namely they add noise to the amplitude of a square pulse. As far as I can tell there's no variability in the time at which the square pulse drops to zero or the time when it rises (let me call such variability "noise in the phase of the square wave"). (If I have misunderstood and their external noise does include this then what is written below can be ignored, but the authors could try to be more clear in their description of the external noise.) Their argument in the dynamical systems section makes me think that such noise in the phase of the square wave would cause the limit cycle oscillators to diffuse in the direction tangent to the trajectory, just as they do with internal noise. But I would still call this external noise because it doesn't have a 1/sqrt(N) dependence. So is the distinction really between external and internal noise (which I interpret as noise that is independent of N vs. 1/sqrt(N)), or is it between noise that affects the phase and noise that doesn't, irrespective of whether it has a 1/sqrt(N) dependence? In the circadian case, maybe there is an argument to say that this is just semantics, because external noise due to weather fluctuations won't change the phase of the day-night cycle. I agree with this, but I'm looking at their results beyond circadian clocks alone, because the audience for their paper will include biologists interested in other oscillators too. For instance, NF-κB oscillations may well be driven by periodic cytokine secretions and "external noise" might well perturb the phase as well as the amplitude of the oscillating cytokine. In such a case, if the real difference in behaviour between limit cycles and hourglass clocks is actually in their response to noise that affects the phase of the driving signal vs noise that does not, then the nomenclature external vs internal noise will inadvertently mislead people. I think simply looking at the behaviour of their phenomenological oscillators with noise in the phase of the square wave will clear this up.

The reviewer brings up an important point here about generalizing our results to non-clock systems. The distinction between external noise and internal noise is not their dependence on the molecular copy number, even though in this circadian clock context, such a description does apply.

We have now clarified the distinction in the Discussion section,

“[...] while the internal noise discussed here is set by finite copy number, this dependence is not essential to the results here. Any source of disturbance (e.g., bursty transcription) that perturbs the phase of the oscillator in constant light conditions is equivalent to internal noise. Similarly, external noise can involve any kind of fluctuation (e.g., multiplicative fluctuations, phase fluctuations) of the external signal that does not result in a persistent phase shift of the external signal itself.”

So indeed, clock architectures that protect against amplitude fluctuations also protect against the kind of phase fluctuations the reviewer describes. However, we are reluctant to discuss this further in the clock context since it is hard to imagine such fluctuations in the day-night cycle of light. It would be interesting to consider the effects of such noise in oscillators like NF-κB.

3) The authors have come up with some neat measures to compare the oscillators they look at. In particular, I think the precision measure (first shown in Figure 1D) based on mutual information will prove to be very useful for other researchers in a variety of contexts. Having a measure like this is critical for this study because otherwise how do you compare different models with different parameter values, different variables, and perhaps even different dimensions? So I actually think the authors undersell themselves here – the measure and its purpose deserves more than one throwaway line in the main text (I doubt anyone will get anything out of "variance along the most informative directions"). If the authors don't want to bring the mathematics into the main text, at least they should say more about comparing models when they talk about Figure 1D (incidentally, although it's clear where they talk about Figure 1D in the main text, an explicit reference to "Figure 1D" is missing and may be useful for a reader who has looked at that figure first and is trying later to find where it is discussed in the text.)

We have now:

1) Added a pointer to a longer discussion of this issue in the Appendix;

2) Added more to the discussion of Figure 1D and cited Figure 1D where we now say:

“This is shown in Figure 1D, where the precision measures the mutual information between the clock state and the time.”;

3) We also discuss this issue later where we say:

“For a fair comparison, in undriven conditions, different clocks are assumed to lose information at the same rate.”

4) In their dynamical systems models, the limit cycle has a constant amplitude. I'm curious: how much would their argument be affected if the radius R in fact changed a lot over the course of the trajectory? (again I'm thinking of very spiky oscillations, or more complex oscillations with non-planar phase plots, which may not be relevant for circadian systems but may be relevant for other biological oscillations.) Wouldn't a varying R cause a non-zero curvature for the limit cycle? If so, then the timing of the day-night transition and whether it happens where curvature is zero or not may become important. How would a large variation of internal noise along the trajectory, due say to R becoming close to zero, affect their results? I'll leave it to the authors to decide whether any of this is in the scope of the current paper or not, but perhaps a quick discussion of the approximation involved in keeping R fixed on the trajectory is warranted.

The reviewer is raising an important question about the shape of the limit cycle attractor. While our dynamical systems theory assumes idealized circular limit cycles, the explicit networks considered in Figure 2 have deformed non-planar limit cycles of the type alluded to by the reviewer. The trade-off continues to hold qualitatively for such cycles.

We have noted this in the Discussion:

“While our dynamical systems theory involve planar circular limit cycles, the models in Figure 2 have complex non-planar non-circular limit cycles and yet reproduce our trade-off.”

For completeness, we note that changing the shape of the limit cycle does not affect the curvature of dynamics along the limit cycle; even deformed limit cycles are ‘flat’ along the limit cycle direction.

5) Whenever the authors talk of the hourglass clocks they present them as having damped oscillations. From their arguments, surely this is not necessary and the system would work as well or as badly if it was not spiraling into the fixed point (eigenvalues having an imaginary component), but rather had an "overdamped" approach (real, negative eigenvalues)? As far as I can tell, as long as the relaxation time is of the same order as the day-night cycle the behaviour should be the same.

The reviewer is mathematically correct. However, if the damped oscillation is over-damped without spiraling, the trajectory a system takes during the day and during the night overlap with each other, and then the clock cannot tell “AM” from “PM”. In fact, the known biochemistry of KaiC, with two distinct kinds of phosphorylation sites, suggests that the two relaxation dynamics between day and night and night and day are not identical.

6) I found parts of the manuscript hard to follow. I needed to flip back and forth between the main text and supplement a fair bit, which to me indicates that some of the supplement needs to go into the main text, or parts of the main text need more amplification. I already mentioned the too brief discussion of the precision measure. In the main text, it also took me a while to figure out what s was (just before Equation 1) – it would help to define this more clearly. And it would help to call out specific sections of the supplement instead of using the generic "see SI". Within the supplement, why not just write out the Langevin equations soon after Appendix 3 Equation 14 rather than asking the reader to jump to find it? And I also think it would be good to include the diffusion matrix the authors have used for each of the "Other oscillators", so that it is clear that they have not used the approximation in Appendix 6 Equation 42 for these simulations (or have they?).

We thank the reviewer for pointing out these presentation issues.

1) We have now expanded on the precision measure as noted earlier.

2) We edited the text around the equations to explain symbols clearly.

For example, for the symbol ‘s’, we now say,”[…] *s*^2^ represents the variance drop during a dawn/dusk entrainment. As shown in the Appendix 5 for external noise (and in Figure 5 for internal noise) this factor *s* can be geometrically explained by the slope of the circle map relating the two cycles[…]”

3) The supplementary files have been completely reformatted into multiple numbered Appendices. We cite specific Appendices in the main text in relevant places.

4) We have now added the Langevin equation after Appendix 3 Equation 14 as requested. We have also cleared up and clarified other issues related to our derivation of the Langevin equations in the Appendices.

We thank the reviewer for helping improve the presentation and readability of our paper.

7) Just out of curiosity – a typical argument made for explaining some aspects of the architecture of circadian oscillators lies in other features such as temperature compensation. Have the authors thought about whether limit cycles or hourglasses do better at temperature compensation?

Most work on temperature compensation has been on free running oscillators. We have not thought through similar considerations for the hourglass clock. We agree that this is an interesting question to answer, given the ecological importance of *Prochlorococcus marinus*.

8) There are a few typos:Subsection “Limit cycle clocks and point attractor clocks”, second paragraph: diverge should be diverse.Subsection “Simulations”, fourth paragraph: Langeving should be LangevinAppendix 6 Equation 39 and Appendix 6 Equation 41: dN_a_ should be dNA.

We fixed these typos.

Reviewer 3:My only concern is with regards to very dense, technical and somewhat cumbersome presentation of the Results. This could be an example where the principle "less is more" could be applicable. The main and most important idea of the paper is clear from Figure 1 and the universal character of the discovered trend shown in Figure 2 is impressive. However, the subsequent analysis in the main text it is very hard to follow and requires full immersion to an appendix to understand. This could be more appropriate for a more specialized journal. Results about trade-off in Figure 6 seems very interesting but presented in a brief passing here.

We appreciate the feedback about. We have taken several steps to improve readability:

1) We broke down the text into multiple themed Appendixes, and cite each Appendix respectively in the main text.

2) We edited the main text to make it more self-contained, removing several references to appendices.

3) We have added more explanations of the dynamical systems theory in the main text and better defined symbols in those equations.

4) We edited the description of the KaiABC model (presented in Figure 1) in the Appendix to better connect it to the dynamical systems theory.

We thank the reviewer for constructive feedback on improving the paper’s presentation.